# Many-objective African vulture optimization algorithm: A novel approach for many-objective problems

Heba Askr[1,2], M. A. Farag[2,3], Aboul Ella Hassanien[2,4]*, Václav Snášel[5], Tamer Ahmed Farrag[6]

**1** Faculty of Computers and Artificial Intelligence, Information Systems Department, University of Sadat City, Sadat City, Egypt, **2** Scientific Research Group in Egypt (SRGE), Nasr City, Egypt, **3** Faculty of Engineering, Department of Basic Engineering Science, Menoufia University, Shebin El-Kom, Egypt, **4** Faculty of Computers and Artificial Intelligence, Cairo University, Giza, Egypt, **5** Faculty of Electrical Engineering and Computer Science, VŠB-Technical University of Ostrava, Poruba-Ostrava, Czech Republic, **6** Department of Computer Engineering, MISR Higher Institute for Engineering and Technology, Mansoura, Egypt

* aboitcairo@cu.edu.eg

## Abstract

Several optimization problems can be abstracted into many-objective optimization problems (MaOPs). The key to solving MaOPs is designing an effective algorithm to balance the exploration and exploitation issues. This paper proposes a novel many-objective African vulture optimization algorithm (MaAVOA) that simulating the African vultures' foraging and navigation behaviours to solve the MaOPs. MaAVOA is an updated version of the African Vulture Optimization Algorithm (AVOA), which was recently proposed to solve the MaOPs. A new social leader vulture for the selection process is introduced and integrated into the proposed model. In addition, an environmental selection mechanism based on the alternative pool is adapted to improve the selection process to maintain diversity for approximating different parts of the whole Pareto Front (PF). The best-nondominated solutions are saved in an external Archive based on the Fitness Assignment Method (FAM) during the population evolution. FAM is based on a convergence measure that promotes convergence and a density measure that promotes variety. Also, a Reproduction of Archive Solutions (RAS) procedure is developed to improve the quality of archiving solutions. RAS has been designed to help reach out to the missing areas of the PF that the vultures easily miss. Two experiments are conducted to verify and validate the suggested MaAVOA's performance efficacy. First, MaAVOA was applied to the DTLZ functions, and its performance was compared to that of several popular many-objective algorithms and according to the results, MaAVOA outperforms the competitor algorithms in terms of inverted generational distance and hypervolume performance measures and has a beneficial adaptation ability in terms of both convergence and diversity performance measures. Also, statistical tests are implemented to demonstrate the suggested algorithm's statistical relevance. Second, MaAVOA has been applied to solve two real-life constrained engineering MaOPs applications, namely, the series-parallel system and overspeed protection for gas turbine problems. The experiments show that the suggested algorithm can tackle

**Data Availability Statement:** The data used in this paper within the text.

**Funding:** This work was supported by the Ministry of Education, Youth and Sports of the Czech Republic in the project Metaheuristics Framework for Multi-Objective Combinatorial Optimization Problems (META MOCOP), reg.no. LTAIN19176.

**Competing interests:** The authors have declared that no competing interests exist.

many-objective real-world applications and provide promising choices for decision-makers.

# 1. Introduction

MaOPs are optimization problems with more than three objectives that must be solved simultaneously [1]. Most real-world applications may have more than four conflicting objective functions and are mathematically being modelled as MaOPs. Some of these applications include automotive engineering, aerospace engineering, many-objective simplified nurse scheduling problem, the five-objective water resource management problem, the ten-objective general aviation aircraft design problem, the many-objective space trajectory design problem, many-objective software refactoring, the hybrid car controller optimization problem with six objectives, optimization of three centrifugal design problems having six to nine objectives, the many-objective 0/1 knapsack problem, Heuristic Learning, Travelling Salesman Problem (TSP), Job shop scheduling, flight control system, supersonic wing design, six-objective design of a factory-shed truss [2], Big data applications which need sophisticated architectures with inherent capabilities to be scaled and optimized [3], NP-hard workflow allocation problems in cloud systems [4], Multicore computers are transforming the embedded computing market [5], and recently Internet of Everything (IoE) [6]. The difficulty of the MaOPs returns to the increase in the problem scale; as the number of objectives grows, the number of nondominated solutions grows exponentially [1]. Solving MaOPs is more difficult for several reasons: the high computational cost of PF approximation due to increased evaluation of several points, the inability of existing evolutionary multi-objective algorithms to solve MaOPs, and the difficulty of visualizing the PF with more than four objectives [2].

The difficulties that Multi-objective Evolutionary Algorithms (MOEAs) experience in solving MaOPs have raised the demand for the development and the deployment of evolutionary algorithms for MaOPs. MOEAs are not scalable enough and have problems addressing the MaOPs. These problems are summarized by [11] as follows: (1) As the number of objective functions grows, the obtained results become non-dominated; (2) As the size of the objective space grows, the conflict between diversity and convergence grows; (3) For computational efficiency, the population size can be small; (4) Computational complexity grows exponentially as the number of objectives grows (for example, hypervolume calculation); (5) Balancing diversity and convergence becomes more complicated; and (6) Due to the vast dimensions, visualizing the Pareto-optimal front is difficult. Due to these challenging issues, MaOPs are more complex and need to be handled using more effective and scalable evolutionary algorithms.

Various ways to solve MaOPs have been proposed as the MOEAs community pays more attention. These approaches can be roughly divided into four categories [2].

## 1.1. Decomposition-based approaches

These non-Pareto-based methods combine the objectives into a scalar function. The weight vector is a weighted coefficient that represents the relevance of each objective. A MaOP is split into numerous single-objective sub-problems that can be optimized simultaneously using a set of weighting vectors.

Scalarization techniques also balance the diversity and convergence of solutions in the objective space. For dealing with MaOPs, [1] presented a new reference direction-based density estimator, a new FAM, and new environmental selection algorithms. To increase the diversity of decomposition-based Evolutionary Algorithm (EA) [7], adopted a dynamical

decomposition technique. Reference vectors were employed by [8] to break down the original MaOP into several single objective subproblems and clarify user preferences to target a preferred subset of the entire PF. The reference points are automatically selected from the solutions and matched to the PF pattern. As a result, these reference points might provide a diversified range of possibilities for guiding the population to explore new areas.

Recently [9], suggested an adaptive decomposition EA (MaOEA/ADEI) based on environmental information. The ecological information determines the penalty factor of Penalty boundary intersection decomposition and includes population and weight vector distribution information. In addition, the weight vectors adaptation approach is employed when dealing with problems involving scaled targets.

## 1.2. The indicator-based approach

The value of the performance indicator is used to direct the search process in this approach. The algorithm in this category used the performance indicator instead of fitness to select individuals. For example [10], introduced a hypervolume estimation algorithm. The exact Hypervolume Values (HV) were approximated using Monte Carlo simulation, and the solutions were rated using the HV indication. An indicator based MOEA with reference point adaption (ARMOEA) was presented by [11]. For MaOPs [12], presented a two-stage R2 indicator-based EA (TS-R2EA). The primary selection is based on an R2 indicator-based achievement scalarizing function. After that, the reference vector guided objective space partition approach is applied as the second selection strategy. A two-stage selection technique yield a good mix of convergence and diversity. In addition, several efficient and effective indicators based MOEAs [13–15] have been presented in the context of these performance metrics.

## 1.3. Pareto-dominance approach

Is the most popular class of MaOPs. Some improved Pareto rank solutions are chosen using dominance-based selection criteria in these approaches. In addition, a diversity-related method will be used to ensure that the Pareto optimal solutions are distributed evenly. Grid domination and grid difference were utilized to strengthen the selection pressure in the authors' [16] Grid-based many-objective evolutionary algorithm (GREA). In addition, to introduce a fuzzy mechanism to Pareto dominance, the authors in [17] employed a continuous function to quantify the degree of non-dominance between two solutions. As a result, solutions with a higher non-dominance degree can be selected. In addition, a novel dominance relation [18] and a reinforced dominance relation [19] were presented to classify just more precisely the best convergent solutions as non-dominated, hence speeding up population convergence. In addition, various efficient and effective Pareto-dominance strategies [20–22] for solving MaOPs have recently been published.

## 1.4. Preference-based approach

This category has three types: a priori, interactive, and posterior. The preference information is supplied before the search in an a priori class. The decision-maker is expected to offer preference information interactively in an interactive class. Similarly, the preference information is introduced after the search in the a posteriori class. Several efficient and effective preference-based EA approaches [23–25] have been proposed to solve MaOPs.

The authors in [26] presented a new nature-inspired metaheuristic algorithm called AVOA in 2021, and it has since been used in several real-world engineering applications. AVOA was created to simulate and model African vultures' foraging behaviour and living habits. Compared to state-of-the-art optimization techniques, the AVOA was determined to be very

promising and powerful. In addition, this technique is substantially faster than any comparable algorithms in terms of computational complexity and running time, and it works well in large-scale applications. The population of African vultures is divided into three groups based on their habits. The first group is to find the best feasible solution among all vultures. The second group is to find the second-best feasible solution among all vultures. The final group is made up of the surviving vultures. The rationale for the division is that each group of vultures has a different ability to locate and consume food. It is assumed that the worst vultures are the weakest and hungriest vultures, and the best vultures are the strongest and most abundant vultures at present. The strongest and best vultures are two of the best solutions in AVOA, while the other vultures are trying to approach the best.

This paper presents a modified version of AVOA to handle MaOPs. This version is called MaAVOA. The AVOA required two best vultures to guide the other vultures to reach the best solution. A new selection process for the MaOPs is introduced and integrated into the proposed model. In addition, an environmental selection mechanism based on the alternative pool is adapted to improve the selection pressure to maintain diversity for approximating different parts of the whole PF. Also, an external Archive based on the FAM is set up to keep track of the best non-dominant solutions as the population evolves. The FAM is based on a convergence measure that promotes convergence and a density measure that promotes variety. Furthermore, a RAS procedure is developed to improve the quality of archiving solutions. The RAS procedure helps to reach out to the missing areas of the PF that the vultures easily miss.

The main contributions of this paper are summarized as follow:

- The proposed MaAVOA is a novel algorithm to solve many objectives problems which achieves promising solutions that promotes diversity and fast convergence.

- The proposed MaAVOA is compared to certain current five best-practice algorithms and achieves results superiority over them, including a unified evolutionary optimization algorithm (U-NSGAIII) [27], a reference-point-based many-objective evolutionary algorithm based on NSGA-II (NSGA-III) [28], A multi-objective evolutionary algorithm based on decomposition (MOEA/D) [29], constrained two-archive evolutionary algorithm (CTAEA) [30], and AGEMOEA adaptive geometry estimation based MOEA (AGE-MOEA) [31].

- The performance of the proposed MaAVOA was evaluated using benchmark functions for DTLZ test suites with some objectives ranging from three to fifteen objectives.

- In addition, it was applied on two real life engineering applications to validate its performance to tackle many-objective real-world applications.

The rest of the paper is organized as follows. The MaOPs and AVOA are presented in Section 2. The proposed algorithm's framework is illustrated in Section 3. The proposed framework's implementation methodology is presented, and the results are discussed in Section 4. In section 5, two engineering applications are introduced. The paper's conclusions and future research initiatives are presented in Section 6.

## 2. Preliminaries

### 2.1. Many-objective optimization problem

Many-objective optimization problems (MaOPs) can be stated as follows:

$$\text{Minimize } F(x) = (f_1(x), f_2(x), \ldots, f_m(x))^T \tag{1}$$

Subjected to

$$Q_g(x) \geq 0, \quad g = 1, 2, \ldots, G$$

$$E_l(x) = 0, \quad l = 1, 2, \ldots, L$$

Where $F: \Omega \rightarrow R^m$ is a set of $m$ conflicting objective functions in the form of a vector, $(m \geq 4)$, $\Omega = \prod_{i=1}^{n} [lb_i, ub_i] \subseteq R^n$ is n-dimensional decision space, $x = (x_1, x_2, \ldots, x_n) \in \Omega$ is a vector of $n$ decision variables (candidate solutions) and $R^m$ is called the objective space [27].

**Definition 1**. (Pareto-dominance) A solution $x^p$ is considered to dominate another solution $x^q$ ($x^p \succ x^q$) if and only if

$$(\forall k \in \{1, 2, 3, \ldots, m\} : f_k(x^p) \leq f_k(x^q)) \wedge (\exists k \in \{1, 2, 3, \ldots, m\} : f_k(x^p) < f_k(x^q)) \quad (2)$$

**Definition 2.** (Pareto-optimal)
A solution $x^p$ is assigned to be Pareto optimal iff: $\exists x^p \in \Omega: x^p \succ x^q$

**Definition 3.** ((Pareto-optimal set (POS)): the set of non-dominant solutions POS includes all solutions that balance the objectives in a unique and optimum manner.

$$POS = \{x^p | \neg \exists x^q \in \Omega : x^p \succ x^q\} \quad (3)$$

**Definition 4**. ((Pareto-optimal front (POF)): The values of all the objective functions corresponding to the Pareto-optimal solutions in POS are included in the set POF.

$$POF = \{(f_1(x), f_2(x), \ldots, f_m(x))^T | x \in POS\} \quad (4)$$

The dimension of the POF is expected to be $m-1$, and the POF is becoming more complex with increasing the number of objective functions, which is the challenge of many-objective optimization problems [7].

## 2.2. Standard African vulture's optimization algorithm

Authors in [26] introduced a novel nature-inspired metaheuristic algorithm, AVOA, used to solve several engineering applications [32]. AVOA was developed by simulating and modelling African vultures' foraging behaviour and living habits.

To simulate the AVOA biological life, four assumptions are considered:

- In the African vulture population, there are $Npop_F$ vultures. Each vulture's position is n-dimensional, with a maximum number of iterations (MaxIter). $X_i^t = [x_1^t, x_2^t, \ldots, x_n^t]$ can be used to indicate the position of each vulture $i$ ($1 \leq i \leq Npop_F$) at different iterations $t$ ($1 \leq t \leq$ MaxIter).

- The population of African vultures is classified into three groups based on their life habits. The first group is to find the best feasible solution among all vultures. The second group is to find the second-best feasible solution among all vultures. The final group is made up of the surviving vultures.

- The division is that each group of vultures has a unique incapacity to discover and eat food.

- The worst vultures are thought to be the weakest and most hungry, while the best vultures are the strongest and most numerous. The strongest and best vultures in AVOA are two of the best solutions, and the other vultures aim to approach the best.

**Table 1. Parameters of the proposed algorithm.**

| Parameter | Description |
|---|---|
| popsize | The population size |
| MaxIter | The maximum number of iterations. |
| *ub, lb* | The upper and lower bounds of the solutions. |
| *dim* | The dimension of the solutions. |
| $X_v^t = [x_1^t, x_2^t, \ldots, x_n^t]$ | The position of each vulture $v$ ($1 \leq v \leq Npop_F$), ($1 \leq n \leq$dim). |
| nRef | Total number of reference points. |
| RP | the set of reference points. |
| $Nr_1$ and $Nr_2$ | The parameters controlling *nRef* along the boundary and inside of the Pareto optimal front. |
| FSLV | The set of the first social leader vultures. |
| SSLV | The set of the second-social leader vultures. |
| k | This parameter denotes the likelihood of the vulture carrying out the exploitation stage. |

## 3. The proposed many-objective African vulture optimization algorithm (MaAVOA)

This paper presents a modified version of AVOA to handle MaOPs. This version is called MaAVOA. Initially, $Npop_F$ Vultures are randomly generated in the decision space using a uniform distribution. After that, vultures are evaluated according to the fitness functions, and the nondominated solutions are identified according to Pareto dominance of NSGA-III [31] then stored in the external archive (ARC). The ARC is based on the FAM, created to keep track of the best solutions as the population evolved. The AVOA required two best vultures to guide the other vultures to reach the best solution. The proposed algorithm uses a set of social leader vultures to guide solutions in the search space. Some of these social leader vultures are chosen from the ARC to lead the other vultures in the population. The proposed algorithm uses FAM in [33], focusing on convergence and diversity to select the first-social leader vultures from ARC. FAM was employed with two objectives to enforce these potential leaders' preferences and learn more about them. MaAVOA iteratively performs a series of steps, the most important of which are (1) obtaining the social leaders for the vultures and moving the solutions in the decision space by using AVOA; (2) Applying polynomial mutation to 10% of the vulture position (candidate solution) to enhance the diversity while avoiding the premature convergence; (3) perform the environmental selection by using the alternative pool to select the best $Npop_F$ vultures for the next generation and (4) Update the external archive to contain only the non-dominated solutions; i,e based on the dominance relation on all objectives. The nondominated solutions in the alternative pool and the old archive are stored in the archive. These steps are repeated up to MaxIter is reached. The parameter of the proposed algorithm is shown in Table 1. The MaAVOA framework is shown in Fig 1 and Algorithm (1) and they are being explained in greater depth in the following subsections.

```
Algorithm 1: MaAVOA
Input: population size Npop_F, MaxIter, and the related parameters.
Output: The position of best vultures and their fitness value
Processing:
  Initialize a random population of vultures X_v(v = 1,2,...,Npop_F)
  Use Pareto dominance of NSG-III to identify non-dominated solutions.
  Save all non-dominated individuals in the archive (ARC)
  While (stopping criteria are not met) do
  • For v = 1: Npop_F
  • Select social leader Vultures (Algorithm 4)
```

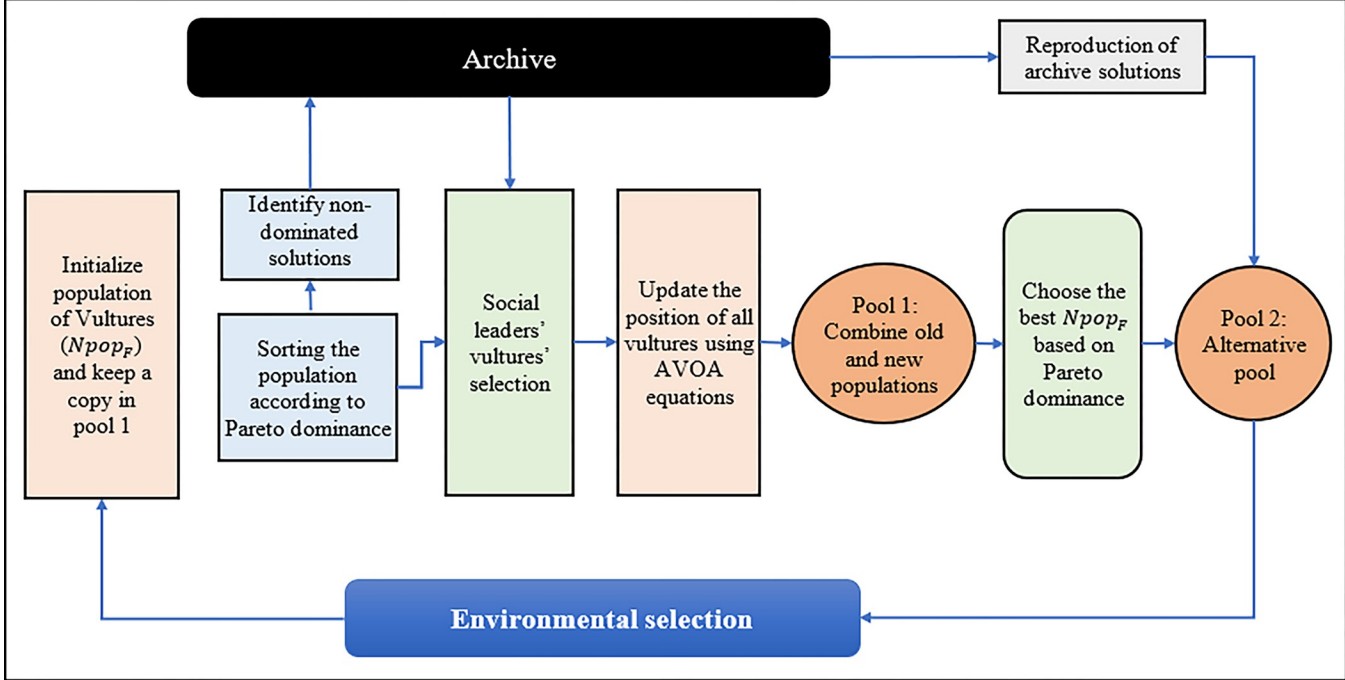

**Fig 1. MaAVOA framework.**

- *pop* = the new position of vultures after updating their position by AVOA
- *pop* = polynomial mutation to 10% of *pop*
- Evaluate the objective values for each individual in *pop*
- Combine the old and new offspring populations, denoted as *px* = *pop*∪$X_v$
- *pop* = sorting *px* by a non-dominated sorting technique of NSGA-III and choosing $Npop_F$ solutions from *px*
- *pop* = Environment Selection from the alternative pool (Algorithm 5)
- Update the ARC by the nondominated solution in the alternative pool.

 **end while**

 return the position of best vultures and their fitness value from ARC

## 3.1. Fitness Assignment Method (FAM)

The MaAVOA's FAM is presented in Algorithm (2) and is based on a convergence measure that promotes convergence and a density measure that promotes variety. This method used a set of reference points to calculate both metrics. These points are utilized to cluster the solutions and, as a result, estimate their density in the objective space. These points are also used to push solutions that are near to the PF.

 MaAVOA uses a collection of reference points to find well-distributed solutions and near the PF. A method for obtaining this set of points was proposed by [34]. This process produces a set of evenly spaced reference points on a hyperplane in the objective space. This hyperplane is in the first quadrant and intersects each axis equally. At position one on each axis, the interception is considered, followed by $Nr$ divisions. As a result, $(nRef = C_{Nr}^{m+Nr-1})$ gives the total number of reference points $nRef$.

```
Algorithm 2: FAM
Input: a population of vultures X_v (v = 1,2,...,Npop_F)
Output: the convergence and density measures
Processing:
```
• Calculate the fitness vector of each vulture.
• Determine the set of reference points $RP = \{rp_1, rp_2,...,rp_{nRef}\}$
• Compute the approximated ideal point $P^{ideal}$
• Compute the new extreme points $Z_t^{extreme}$ from $Z_{t-1}^{extreme} \cup$ ARC
• Compute the hyperplane from extreme points.
• Compute the density measure and the convergence measure of each solution

   The basic steps for calculating the density measure and the convergence of the solutions in ARC are illustrated in the following steps.

- **Step1**: Generate a set of reference points $RP = \{rp_1, rp_2,...,rp_{nRef}\}$ by using a method proposed by [33], where *nRef* is the total number of reference points. For example, if m = 4 objective functions, the reference points are created on a rectangle with apex at (1, 0, 0,0), (0, 1, 0, 0), (0, 0, 1,0) and (0, 0, 0,1) with considering four divisions (*Nr* = 4), and 35 reference points will be generated.

- **Step 2:** identify ideal point $P^{ideal} = (f_1^{min}, f_2^{min}, \ldots, f_m^{min})$, then each objective function for each solution in ARC is transformed to $\tilde{F}$ by subtracting the value in objective $F(X)$ by $P^{ideal}$, i.e. the translated objective *i* is obtained from $f_i^{\sim} = f_i - f_i^{min}$.

- **Step 3:** Compute the set of the extreme solutions $Z_t^{extreme}$ from all solutions in ARC up to the current iteration (t) of the algorithm. The solution $i \epsilon (ARC \cup Z_{t-1}^{extreme})$ is an extreme solution for objective *n* if this solution *i* minimize the scalarizing achievement function (AS) as follows.

$$AS(x_i, rp_n) = max_{j=1}^{m} \frac{f_j^{\sim}(x)}{rp_{nj}} \qquad \forall i \epsilon (ARC \cup Z_{t-1}^{extreme}) \qquad (5)$$

where $rp_n = \{rp_{n1}, rp_{n2},...,rp_{nm}\}$ is a unitary vector that corresponds to the direction in the axis n, that is, $rp_{nj} = 0$ if $n \neq j$ and $rp_{nk} = 1$ otherwise in which $n \in \{1, 2, \ldots, m\}$. Using this method, all the solutions found in the ARC thus far are used to update the set $Z_t^{extreme}$ extreme solutions. Then, the *m* objective vectors of the in $Z_t^{extreme}$ are used to build a hyperplane in the objective space and extended to reach these *m* objective vectors. The intercept $d_i$ of the i-th objective axis and the linear hyperplane can then be obtained by calculating the distance from the interception point and the origin and using this value to normalize the objective functions.

$$norF_i(x) = \frac{f_i^{\sim}(x)}{d_i - P_i^{ideal}} \qquad (6)$$

- **Step 4:** Associate the solutions in ARC to the reference point. For this purpose, each reference point is joined with the origin to construct a reference line corresponding to each reference point on the hyperplane. The distance perpendicular to each solution in ARC to the reference lines is computed. Each solution is associated with the closest reference point, whose reference line is closest to it in the normalized objective space. As a result, each reference point will have a set of solutions. Now the density around each reference point can be estimated by, counting the number of ARC solutions linked to it. Therefore, the density measure ($Dm_j$) of each solution in ARC is equal to the size of the group in which it is

associated with it. For example, if the solutions $\omega_i = \{x, w, y, z\}$ form the cluster $\omega_i$ of a reference point $rp_i$, then the $Dm_j$ of these solutions is equal to 4.

- **Step 5:** Compute the convergence measure ($conv_j$) to promote convergence. For each solution in ARC, the AS function and associated reference point are calculated, which is the convergence measure for this solution and donated by ($conv_j$). For each reference point $rp$, the AS is calculated of the solutions from the external archive associated with this reference point concerning it (using Eq (5)). Mathematically, the convergence measure of a solution j ($conv_j$) is calculated as follows.

$$conv_j = AS(x_j, rp_i) \quad i \in \omega_i \tag{7}$$

Based on the above four assumptions of AVOA in section 2.2, To simulate the diverse vulture behaviours in the foraging stages, MaAVOA can be divided into five phases. The first phase is the social vulture's selection, the rate of hunger of vultures is the second phase, the exploration and exploitation phases are the third and fourth phases, respectively, and finally, the environmental selection phase is to select the best $Npop_F$ vultures for the next generation. The flowchart for simulating various vulture behaviours in the foraging stages is shown in Fig 2 and presented in more detail in next subsections.

## 3.2. The social leader vultures selection

MaAVOA needs two social leader vultures to guide the other vultures in the population. According to MaOPs, there is no one best solution over the population for the investigated problem. Instead, there are a set of non-dominated solutions, so we will select two sets of social leader vultures. In the proposed MaAVOA, the social leader vultures will be divided into two sets: the first social leader vultures (FSLV) and the second-social leader vultures (SSLV). The FSLV set contains all non-dominated solutions in the ARC. For each vulture in the population, the first social leader is chosen from the FSLV by using the measurements of diversity and convergence in FAM to separate the ARC's solutions such that the best solutions are chosen based on these criteria The tournament selection procedure is used to assign the first social leader vulture ($fsl^v$) from the FSLV to this solution $v$. A solution i $\in$ ARC is better than a solution j $\in$ ARC in the tournament selection procedure if it has a density value lower than the second one. If the two solutions have the same density, the convergence measure determines which is preferable. In the event of a tie, we choose solution $i$ as a leader if $Dm_i < Dm_j$. Otherwise, solution j is selected. Aside from that, solution j is chosen.

For the second-social leader vultures ($SSLV$) set, the guiding vultures, in this case, will be a set of best solutions corresponding to each objective function from all vultures in the population. This selection procedure aims to find the best solutions that are closer to the PF. As a result, each iteration's hyperplane is pushed closer to the PF, improving the convergence. The set of second-social leader vulture position $SSLV = \{sl_1, sl_2, \ldots, sl_m\}$ consists of $m$ best solutions, one for each objective. Thus, each vulture in SSLV is dedicated to bringing the new vultures closer to the PF's ideal point. For selecting the second social leader $ssl^v$ for a vulture $v$, the random selection process assigns this leader from SSLV to this solution.

The selection process of the set of $SSLV$ is given in Algorithm (3), and the social leader vulture selection process ($fsl^v$ and $ssl^v$) for each vulture in the population is given in Algorithm (4).

**Algorithm 3: Second-Social vulture set selection for each vulture**
**Input:** population of vultures
**Output:** *SSLV*
**Processing:**

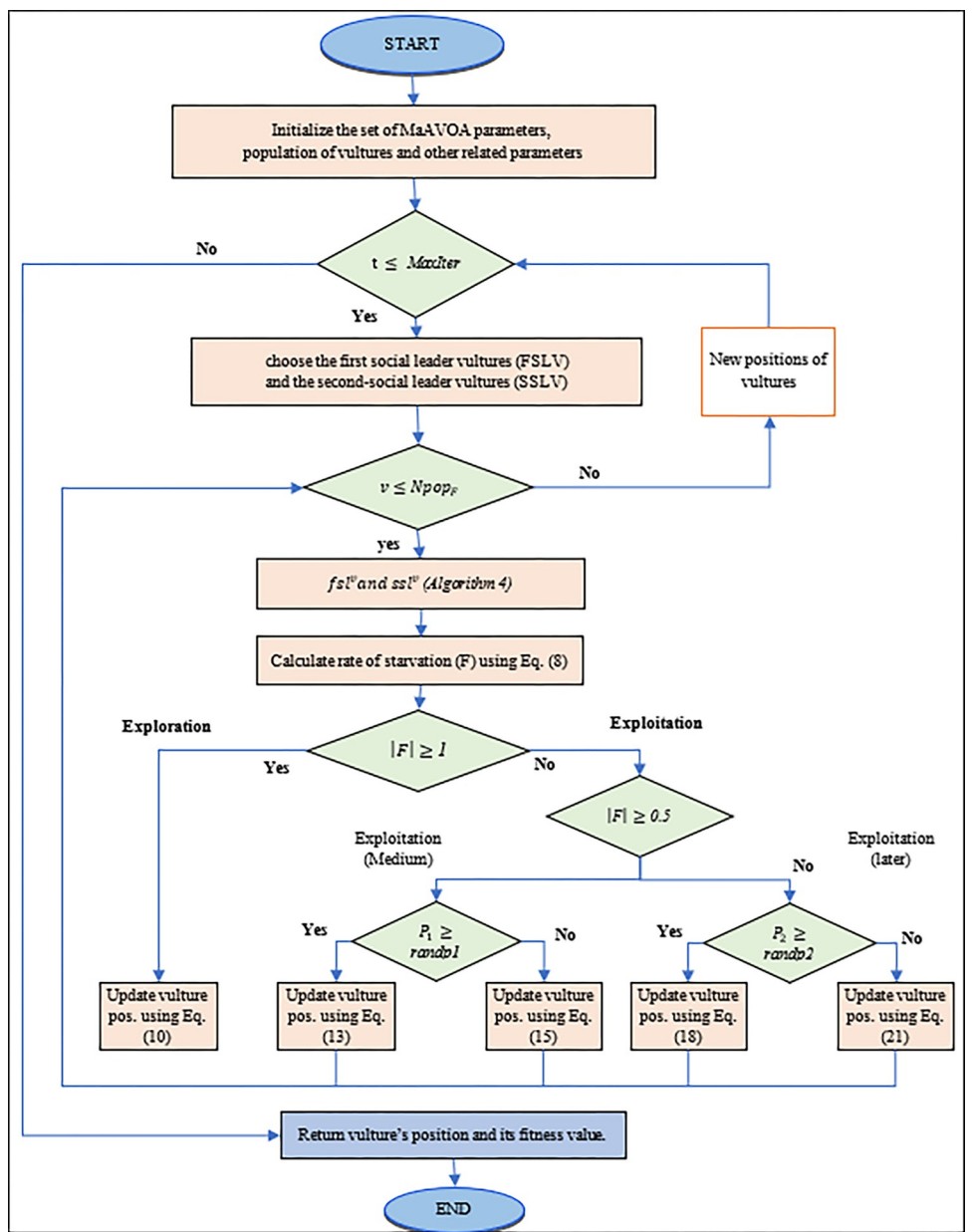

**Fig 2. The flowchart of simulating of various vulture behaviours in the foraging stages.**

- Compute the objective functions for each vulture.
- Assign the minimum objective function for each objective
$$P^{ideal} = (f_1^{min}, f_2^{min}, \dots, f_m^{min})$$
- Define each vulture corresponding to each objective (i.e $v_1$ corresponding to $f_1^{min}$, $v_2$ corresponding to $f_2^{min}$, $v_m$ corresponding to $f_m^{min}$)
- Output $SSLV = \{sl_1, sl_2, \dots, sl_m\}$

**Algorithm 4: Social leaders vultures' selection**

**for** $v$ = 1: $Npop_F$)
  $[ARC(i), ARC(j)]$ = tournament selection ($ARC$)
  Compute the density measure ($DM$) and the convergence measure ($conv$) of each $ARC(i)$ and $ARC(j)$ (Algorithm 2)

```
If Dm_i<Dm_j
   fsl^V = ARC(i)
elseif Dm_j<Dm_i
   fsl^V = ARC(j)
elseif Dm_j = Dm_i
    if con_i<con_j
       fsl^V = ARC(i)
    else
        fsl^V = ARC(i)
    end
end
s = random[1, m]
ssl^V = SSLV(s)
end
```

### 3.3. Vultures' hungry rate

The vulture has the strength to fly to obtain food if it is not hungry. If the vulture is very hungry, it lacks the strength to fly large distances. As a result, hungry vultures will stick near the vultures with food rather than searching for food on their own. The exploration and exploitation stages of vultures can thus be formed based on the above behaviour. The degree of hunger indicates when vultures are transitioning from the exploration to the exploitation stage. The $i^{th}$ vulture's hunger degree $F_i^t$ at the $t^{th}$ iteration can be calculated by

$$F_i^t = \left(2 \times rand_{i1}^t\right) + 1 \times z^t \times \left(1 - \frac{t}{T}\right) + g^t \tag{8}$$

$$g^t = h^t \times \left(\sin^k\left(\frac{\pi}{2} \times \frac{t}{T}\right) + \cos\left(\frac{\pi}{2} \times \frac{t}{T}\right) - 1\right) \tag{9}$$

where $rand_{i1}^t$ is a random number between 0 and 1, $z^t$ is a random number between -1 and 1, $h^t$ is a random number between -2 and 2, and $k$ is a parameter that has been set in advance, this denotes the likelihood of the vulture carrying out the exploitation stage.

When $|F_i^t|$ is greater than 1, vultures enter the exploration stage, searching for new food in various locations. When $|F_i^t|$ is less than 1, vultures enter the exploitation stage, looking for better food in the immediate vicinity.

### 3.4. Exploration stage

Vultures in AVOA can investigate different random locations using two alternative tactics, which are selected using a parameter called $p_1$. This parameter $p_1$ is given with the algorithm's initialization, and the range is [0,1]. The exploration stage of the vulture can be expressed as

$$X_i^{t+1} = \begin{cases} leader_i^t - D_i^t \times F_i^t, & p_1 \geq rand_{p1}^t \\ R_i^t - F_i^t + rand_{i2}^t \times ((ub - lb) \times rand_{i3}^t + lb), & p_1 < rand_{p1}^t \end{cases} \tag{10}$$

$$leader_i^t = \begin{cases} fsl^i & if \ pr = s1 \\ ssl^i & if \ pr = s2 \end{cases} \tag{11}$$

where $X_i^{t+1}$ is the the $i^{th}$ vulture's position at the $t+1^{th}$ iteration, $rand_{p1}^t$, $rand_{i2}^t$, $rand_{i3}^t$ and $pr$ are random numbers that are uniformly distributed in the range [0,1]. $leader_i^t$ is the social leader vulture as $fsl \in FSLV$ and $ssl \in SSLV$, which are chosen for vulture i in Algorithm (4). s1

and s2 are parameters that were measured in advance, with values ranging from 0 to 1, and the sum of both being one. $F_i^t$ is calculated according to Eq (8), $ub$, $lb$ represent the upper and lower bounds of the solutions, and $D_i^t$ represents the distance between the vulture and the current optimal vulture and calculated by:

$$D_i^t = |C \times R_i^t \times X_i^t| \tag{12}$$

where $C$ is the vultures move randomly to protect food from other vultures.

## 3.5. Exploitation stage at medium level

If the value of $|F_i^t|$ is less than 1, then AVOA enters the exploitation phase, divided into two phases, each with two alternative methods (medium and later).

**3.5.1. Competition for food.** The weaker vultures try to exhaust the healthier vultures and get food from them by congregating around them and provoking minor confrontations. Based on this behaviour, the vultures' position is updated and the updated formula can be expressed as:

$$X_i^{t+1} = D_i^t \times (F_i^t + rand_{i4}^t) - d_i^t \tag{13}$$

$$d_i^t = leader_i^t - X_i^t \tag{14}$$

**3.5.2. Rotating flight of vultures.** When a vulture is full and active, it will not only compete for food but also hover at high altitudes, according to AVOA's spiral model. The updated formula can be expressed as:

$$X_i^{t+1} = leader_i^t - (S_{i1}^t + S_{i2}^t) \tag{15}$$

$$S_{i1}^t = leader_i^t \times \left( \frac{rand_{i5}^t \times X_i^t}{2\pi} \right) \times \cos X_i^t \tag{16}$$

$$S_{i2}^t = leader_i^t \times \left( \frac{rand_{i6}^t \times X_i^t}{2\pi} \right) \times \cos X_i^t \tag{17}$$

## 3.6. Exploitation stage at later level

When the value $|F_i^t|$ is less than 0.5, almost all vultures in the population were full, but after a long period of time, the best two species of vultures were hungry and feeble. Vultures will attack food at this time, and several different vultures will congregate around the same food source.

**3.6.1. Aggregation behaviour.** Vultures have digested a large portion of the food during the late stages of AVOA. Where there is food, many vultures will congregate, and competition will ensue. At this point, the vulture position update formula is as follows:

$$X_i^{t+1} = \frac{A_{i1}^t + A_{i2}^t}{2} \tag{18}$$

$$A_{i1}^t = fsl_i^t - \frac{fsl_i^t \times X_i^t}{fsl_i^t - (X_i^t)^2} \times F_i^t \tag{19}$$

$$A_{i2}^t = ssl_i^t - \frac{ssl_i^t \times X_i^t}{ssl_i^t - (X_i^t)^2} \times F_i^t \tag{20}$$

**3.6.2. Attack behaviour.** When AVOA is in its last stages, the vulture will flock to the best vulture to scavenge the remaining food. The vultures' position update formula can be expressed at this point as in Eq (21).

$$X_i^{t+1} = leader_i^t - |d_i^t| \times F_i^t \times Levy(dim) \tag{21}$$

where $dim$ represents the solution's dimension, $Levy(dim)$ represents the Levy flight [26], and its calculation formula is given by the following Equation.

$$Levy(dim) = 0.01 \times \frac{r_{1 \times \sigma}}{|r_2|^{\frac{1}{\delta}}} \tag{22}$$

where $r_1$ and $r_1$ are uniformly distributed random numbers in the range [0,1], $\delta$ is a constant, which is usually set to 1.5, and the calculation formula of $\sigma$ is given by nest equation.

$$\sigma = \left( \frac{\Gamma(1+\delta) \times \sin\left(\frac{\pi\delta}{2}\right)}{\Gamma(1+\delta) \times \delta \times 2^{\left(\frac{\delta-1}{2}\right)}} \right)^{\frac{1}{\delta}} \tag{23}$$

where $\Gamma(x) = (x-1)!$

## 3.7. Environmental selection operator

The ARC stores non-dominated solutions found by the algorithm during the search process until the algorithm is completed. The archive stores the nondominated solutions from all vultures for information sharing. A vulture may have very poor values on some objectives when the number of objectives increases. These poorly performing objectives need the solutions of ARC for information sharing. Thereby the vulture is pushed to converge to the PF. Although the MaAVOA's environmental selection operator has achieved a reasonably balanced performance in terms of convergence and diversity, the MaAVOA's new progeny may have a diversity problem with other solutions. The idea of integrating into the alternate pool is being tested as a solution to this challenge. By integrating the reproduction generated by the genetic operators to the solutions in the ARC to construct the alternative pool containing the new offspring generated by the MaAVOA operator and archive offspring generated by reproduction of archive solutions to select the best $Npop_F$ vultures according to the dominance relation on all objectives. Under the pressure of the alternative pool, the algorithm assures that the operators work together to find more extended alternative solutions in the population's evolutionary process. As a result of the effect of the alternate pool, the algorithm's overall evolutionary efficiency improves. Population convergence and distribution are ensured because of the environmental operator's influence. We used some ideas and schemes from [35] to develop this environmental selection.

Archive solutions (RAS) were reproduced on 50% of the ARC solutions. In RAS, crossover and mutation processes inherit different dimensions from different solutions. Some parents from the archive are selected randomly and then perform simulated binary crossover (SBX) and polynomial mutation (PM), and these new solutions are then added into the alternative pool then choose the best $Npop_F$ individuals according to the dominance relation on all objectives.

```
Algorithm 5: Environment Selection (RAS, pop) Operator.
Input: pop (offspring generated by the MaAVOA), ARC (solutions in
archive)
Output: pop (new generation of vultures).
Processing:
PAS = choose random 50%of vultures from ARC
RAS = Genetic operators to (PAS)
for i = 1 to |RAS|
  for j = 1 to |pop|
    Judge the dominance relation between RAS(i) and pop(j);
    if the nondominated solution is located in pop
      Retain the corresponding nondominated pop solutions;
    end if
    if the nondominated solution lies in RAS
    add the corresponding nondominated RAS(i) 1to pop;
    end if
  end for
end for
if |pop| > Npop
  Compute the fitness values using the FAM method; (Algorithm 2)
  Remove some solutions with the worst fitness values;
end if
Output the pop with size Npop_F for the next generation.
end
```

## 3.8. Updating the external archive

Because the social leader vultures are chosen from the ARC, good administration of this archive is crucial and significantly impacts the algorithm's performance. The external archive is updated at each iteration. We attempt to place each non-dominated solution from the vultures obtained after environmental selection in the external archive. If any archive solution dominates this added solution, it is neglected. Otherwise, this solution is saved to the external archive, and the solutions dominated by this new non-dominated solution are deleted from the archive.

## 4. MaAVOA implementation

Five state-of-the-art algorithms are compared to our proposed algorithm, namely a unified evolutionary optimization algorithm (U-NSGAIII) [27], a reference-point-based many-objective evolutionary algorithm following NSGA-II [28], A multi-objective evolutionary algorithm based on decomposition (MOEA/D) [29], constrained two-archive evolutionary algorithm (CTAEA) [30], and AGE-MOEA adaptive geometry estimation based MOEA (AGE-MOEA) [31]. These algorithms have been developed to solve MaOPs. The proposed and the state-of-the-art algorithms have been implemented and added to the modern Multi-Objective Optimization package (Pymoo). To evaluate the performance of the proposed algorithm, it is applied to both benchmark problems (DTLZ1-DTLZ7) and two engineering application: Series-parallel system problem and Overspeed protection for gas turbine [36] as case studies. Wilcoxon Test Statistic has applied on all the experiments.

All experiments are tested on a machine with the following specifications: CPU: Core i5 Processor 2.5 GHz /16GB RAM /500GB SSD, GPU: NVIDIA GeForce GTX1050 4GB, compute capability 6.1.

### 4.1. Benchmark problems

In the proposed work, we used the DTLZ1-DTLZ7 benchmark problems, and they are commonly used due to their scalability for any number of objective functions. It is a widespread

**Table 2. PF of DTLZ benchmark problems.**

| Benchmark problem | Description |
|---|---|
| DTLZ1 | The optimal Pareto front lies on a linear hyperplane $\sum_{m=1}^{M} f_m = 0.5$ |
| DTLZ2 | The search space is continuous, unimodal and the problem is not deceptive |
| DTLZ3 | The search space is continuous, unimodal and the problem is not deceptive. It is supposed to be harder to converge towards the optimal Pareto front than DTLZ2. |
| DTLZ4 | The search space contains a dense area of solutions next to the $f_m/f_1$ plane. |
| DTLZ5 | This problem will test an MOEA's ability to converge to a curve and will also allow an easier way to visually demonstrate the performance of an MOEA. Since there is a natural bias for solutions close to this Pareto-optimal curve, this problem may be easy for an algorithm to solve. |
| DTLZ6 | A more difficult version of the DTLZ5 problem where the non-linear distance function g makes it harder to convergence against the Pareto optimal curve. |

test suite conceived for MaOPs with scalable fitness dimensions [37]. All the problems in this test set are scalable in the fitness dimension and are continuous n-dimensional many-objective issues. The decision space has a dimension of k + m + 1, where m is the number of objectives, with k = 5 for DTLZ1, k = 10 for DTLZ 2–6, and k = 20 for DTLZ7, as proposed in [37]. Table 2 lists the properties of the decision space and the PF for each problem.

## 4.2. Parameter settings

Concerning the recommended parameter settings for the compared algorithms, crossover and mutation probabilities are set to 1 and 1/D, respectively. The mutation and crossover distribution parameters have been set to 20. The population size of all algorithms is set to be the same to make a fair comparison with other algorithms. Table 3 shows the number of reference points (nRef) for problems with different objectives. We set population size (popsize) equal to nRef for both the state of the art and MaAVOA algorithms. We used the same settings in [38]. One layer of reference points for three- and five-objective problems and two layers of reference points for eight-ten-fifteen-objective problems are used according to [39]. The reference points (or popsize) are set according to parameters $Nr_1$ and $Nr_2$ for the different number of objectives. $Nr_1$ and $Nr_2$ are parameters controlling nRef along the boundary and inside of the Pareto optimal front (used in the previously mentioned calculation of the number of reference points ($nRef = C_{Nr}^{m+Nr_1-1}$).

For a fair comparison, each state-of-the-art algorithm is applied to solve the DTLZ benchmark functions using three different cases or scenarios to analyse the proposed algorithm's performance and discuss its weak points. The first case is terminating the algorithms after 500 generations for each test problem. The second case is terminating the algorithms after 100000 function evaluations for each run. The third case is setting the computational time of each run

**Table 3. Settings of the reference points and population size.**

| objectives | $Nr_1$ | $Nr_2$ | nRef | popsize |
|---|---|---|---|---|
| 3 | 16 | 0 | 153 | 153 |
| 5 | 6 | 0 | 210 | 210 |
| 8 | 3 | 2 | 156 | 156 |
| 10 | 3 | 2 | 275 | 275 |
| 15 | 2 | 1 | 135 | 135 |

to 3 seconds for all the algorithms. In each of the three scenarios, each algorithm is run 20 times separately on each test problem.

## 4.3. Performance indicators

Three widely used performance metrics are utilized to evaluate the performance of algorithms in this paper. Generational Distance (GD), Inverted Generational Distance (IGD), and hypervolume (HV). All of them can be an indicator for the convergence and distribution of a solution set as comprehensive performance measures [40].

- Inverted Generational Distance (IGD) and Generational Distance (GD) are two measurement indicators used to validate the results. The GD performance indicator measures the solution's distance to the PF. Let us assume the points found by our algorithm are the objective vector set $A = \{a_1, a_2,\ldots,a_{|A|}\}$ and the set of evenly sampled solutions from the genuine Pareto optimum front is $Z = \{z_1, z_2,\ldots,z_{|A|}\}$ then,

$$GD(A) = \frac{1}{|A|}\left(\sum_{i=1}^{|A|} d_i^p\right)^{1/p} \tag{24}$$

where $d_i$ represents the Euclidean distance (p = 2) from $a_i$ to its nearest reference point in $Z$. Basically, this result is the average distance from any point $A$ to the closest point in the PF.

- The IGD performance indicator inverts the generational distance and measures the distance from any point in $Z$ to the closest point in $A$.

$$IGD(A) = \frac{1}{|A|}\left(\sum_{i=1}^{|A|} \hat{d}_i^p\right)^{1/p} \tag{25}$$

where $\hat{d}_i$ represents the Euclidean distance (p = 2) from $z_i$ to its nearest reference point in $A$.

- **Hypervolume (HV):** The volume covered by the obtained PF in the object region, defined as the HV between the front surface and the reference vector, is used to represent the volume covered by the obtained PF in the object region. As a result, the HV reflects the PF's solution distribution. To calculate HV, we set the reference point of HV to $p = (1,1,\ldots,1)^T$ [41]. To guarantee that the individuals in the population can contribute to HV as much as possible, the objective values are normalized by 1.1 times the nadir point of the PF. The HV metric is calculated accurately when the number of objectives is less than 5. When the number of objectives is greater than 5, the Monte Carlo method is adopted to calculate HV. We used 106 sample points for a more accurate result.

## 4.4. Results and discussion

This section analyses all the outcomes acquired from various experiments conducted throughout the implementation phase in this paper.

**4.4.1 Convergence analysis.**   The ability of the global search method to converge is a critical performance criterion for MaOPs. This part looked at the MaAVOA's convergence as a function of the number of iterations using the IGD measure. The convergence trajectories have been chosen randomly from 30 algorithm runs of the MaAVOA and the other five algorithms on DTLZ1-4 with three and ten objectives in Fig 3.

In case of DTLZ1 and DTLZ4 with 3 and 10 objectives, all algorithms exhibit a similar and strong ability to converge to PF, except CTAEA and AGE-MOEA have the worst convergence in all test problems. In addition, it is noted that the convergence of the proposed MaAVOA

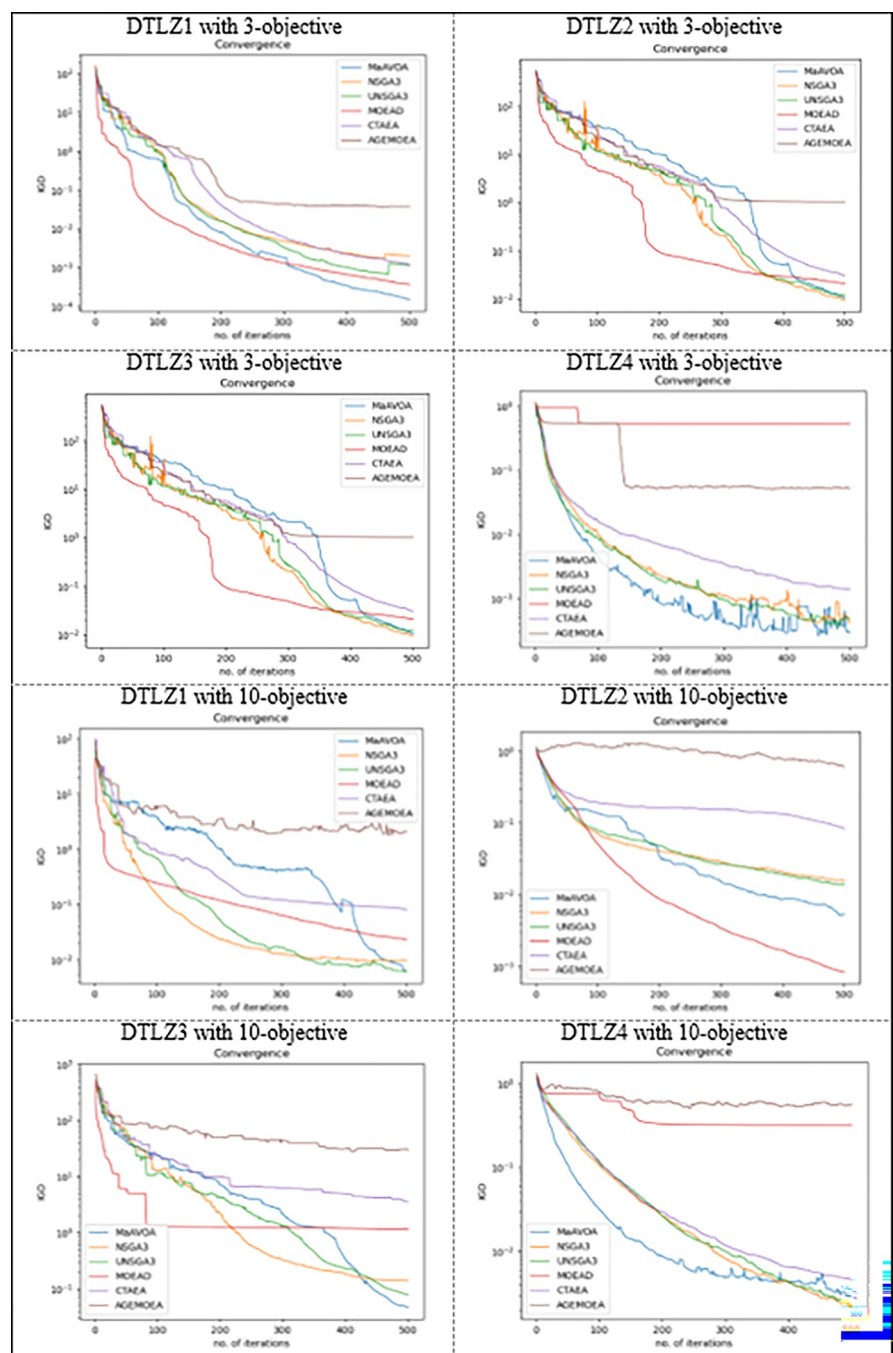

**Fig 3. Convergence trajectories of seven algorithms on DTLZ1-4 with 3 and 10 objectives.**

towards PF is better than the convergence of NSGAIII in most problems. This returns to MaA-VOA uses an external archive where the non-dominated solutions found by the algorithm during the search process are stored through the algorithm. In contrast, NSGA-III worked only on the updated population. The solutions in the external archive are used to lead the other solutions in the population, and the MaAVOA uses FAM with two objectives having the simultaneous goal of imposing preferences among these potential social leaders. The proposed approach uses Pareto dominance and information about density and proximity to push the

vultures towards the PF, which comprises a significant difference between the proposed MaA-VOA and NSGA-III.

Although MaAVOA and U-NSGAIII have approximately the same convergence, MaAVOA is still better in convergence in all the problems except DTLZ with 3 objectives. In addition, MOED/D shows a decrease in convergence in the case of DTLZ1 and DTLZ 4 with 3 objectives and DTLZ 3 and DTLZ 4 with 10 objectives. Furthermore, MaAVOA shows great performance on DTLZ1 with 3 or 10 objectives, DTLZ3 with 3 or 10 objectives, and DTLZ2 and DTLZ 4 with 10 objectives. This demonstrates its great ability to solve MaOPs problems with concave PF. We can observe that the proposed algorithms show terrible performance on DTLZ2 and DTLZ 4 with 3-objectives compared to those with 10 objectives. This returns to MaAVOA new strategy to choose social leader vultures that guide the other vultures to PF. On several tests, it demonstrates good scalability in terms of the number of decision variables and it is concluded that the suggested algorithm has a promising convergence ability to PF.

The other obtained PFs for all DTLZs can be found on https://github.com/tfarrag2000/MaAVOA.We can clearly observe the convergence and diversity of MaAVOA solutions for the high dimensional MaOPs.

In Figs 4 and 5, the approximate PF obtained by the six competing algorithms on DTLZ3 and DTLZ7 with 3,4, and 10 objectives problems is presented to further explain the results.

As shown in Fig 4, NSGA-III, CTAEA, MOEA/D, and MaAVOA have a good distribution, indicating that they performed well on DTLZ3 with three objectives. But U-NSGA-III is unable to maintain convergence and distribution of the solutions. In addition, AGEMOEA failed to converge to the true PF. When the algorithms being tested on DTLZ7 with 3 objectives, MaAVOA and UNSGA-III show superior performance than the other algorithms. It is well observed that the proposed MaAVOA has great diversity and convergence but NSGA-III, CTAEA, AGEMOEA and MOEA/D cannot converge to the true PF on DTLZ7 with 3 objectives.

As shown in Fig 5, NSGA-III, U-NSGA-III, and MaAVOA demonstrate a good dispersion, displaying their excellent performance on DTLZ3 and DTLZ7 with 10 objectives, while CTAEA, AGEMOEA and MOEA/D have a bad ability of convergence and diversity. Figs 4 and 5 show that MaAVOA has shown a good dispersion, demonstrating their superior performance.

**4.4.2 Results for GD, IGD, and hypervolume.** *Case 1: the termination condition is set to be 500 generation*.

The IGD results of the six algorithms on the seven DTLZ tasks with 3,5,8, and 10 objectives are presented in Table 4 while Table 5 shows the values of the GD results of several algorithms and Table 6 shows the values of the HV results as well.

The results of Tables 3 and 4, show that the proposed MaAVOA has achieved competitive performance for most test cases, indicating that MaAVOA can better balance convergence and diversity than the five comparative algorithms.

*Case 2: the termination condition is set to be 100000 function evaluations*.

Tables 7–9 give the values of the three metrics IGD, GD, and HV of the six algorithms on the seven DTLZ problems with 3,5,8 and10 objectives when the termination condition of all algorithms is set to be 100000 function evaluations.

As seen from Tables 7–9, MaAVOA has achieved competitive performance for most test problems.

*Case 3: The termination condition is set to be computational time equal 30 seconds*

Tables 10–12 gives the values of the three metrics IGD, GD, and HV of different algorithms on the seven DTLZ problems with 3,5,8 and10 objectives when the termination condition of all algorithms is set to be 30 seconds. In addition, Table 13 compares the proposed MaAVOA

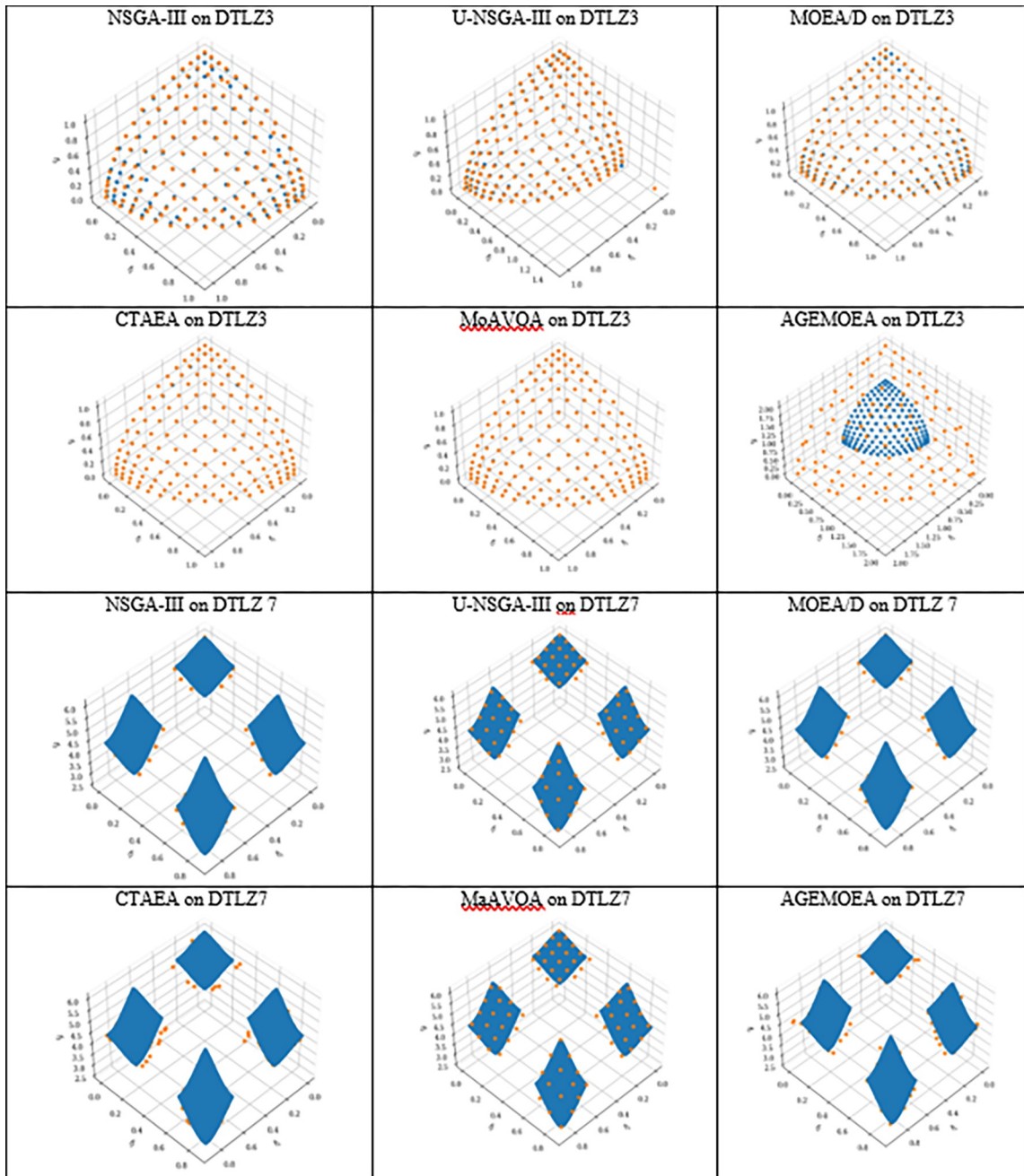

**Fig 4. The parallel coordinates of the non-dominated front obtained by each algorithm (used in the comparison) on DTLZ3 and DTLZ7 with 3 objectives.**

with the other algorithms in terms of the number of generations and number of function evaluations on DTLZs in the case of the computational time being 30 seconds.

As shown in Table 13, when all algorithms end after 30 seconds, the MaAVOA has achieved competitive performance for most test problems, the same as in the other two previous cases. On the other hand, Table 13 shows that the proposed algorithm has implemented for several generations that is smaller than the NSGAIII, U-NSGAIII, and AGEMOEA and also several function evaluations of MaAVOA is smaller than NSGAIII and U-NSGAIII as well for 30 seconds.

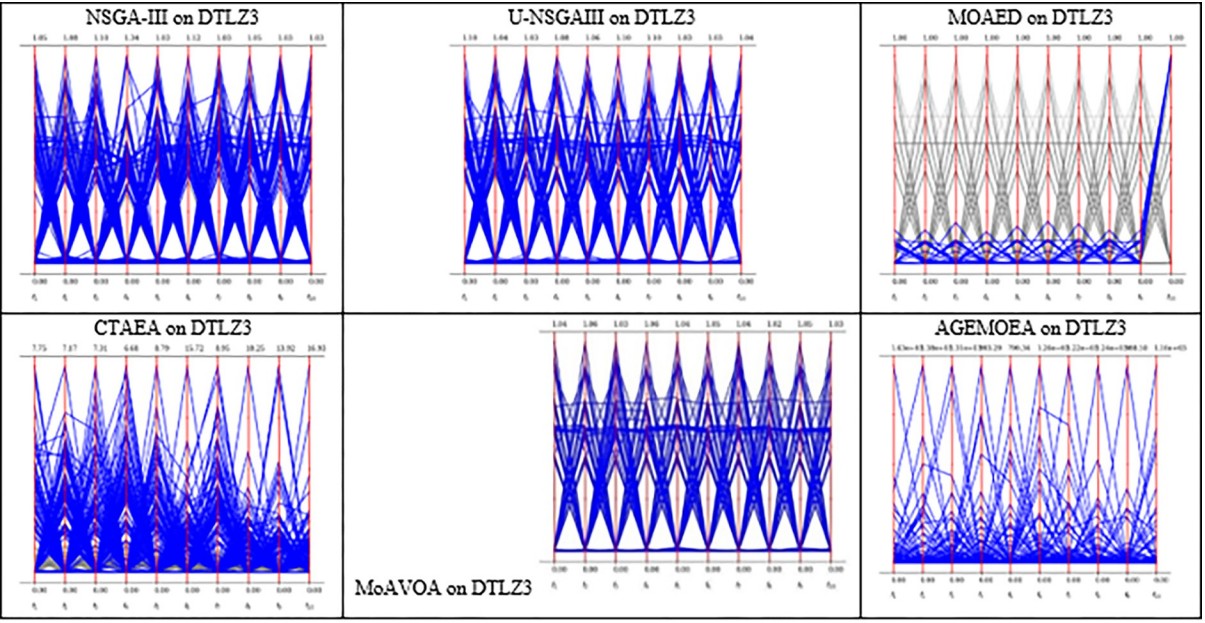

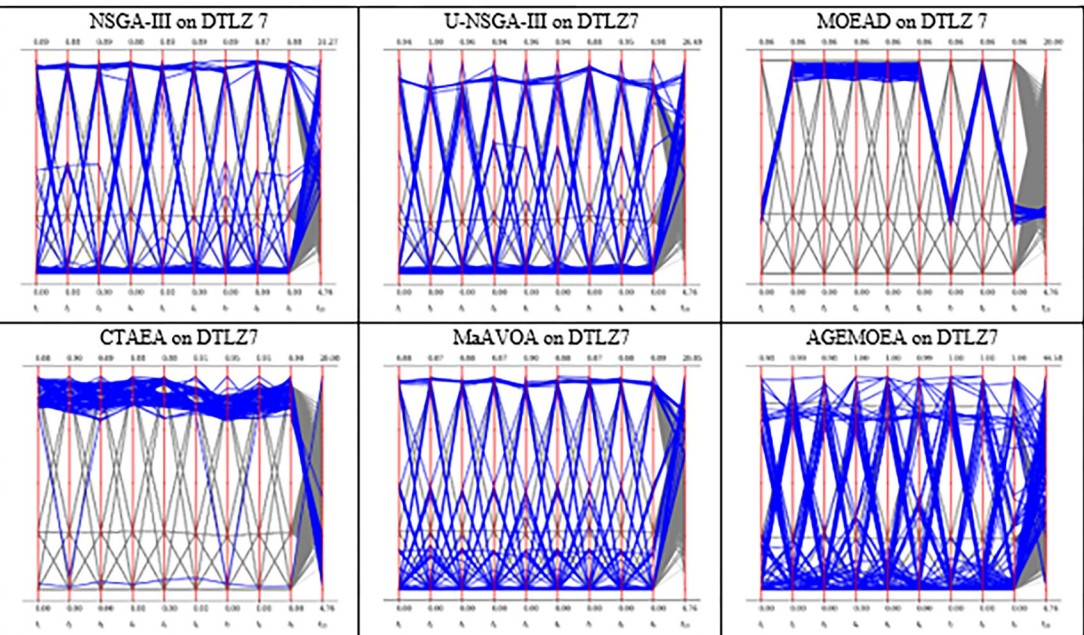

**Fig 5. The parallel coordinates of non-dominated front obtained by each algorithm (used in the comparison) on DTLZ3 and DTLZ 7 or any function with 10 objectives.**

## 5. MaAVOA for engineering applications

In this section, the performance of the proposed MaAVOA has been tested on two real-life engineering applications, namely, the Series-parallel system and Overspeed protection for gas turbine. These applications are used to show the efficiency and effectiveness of the proposed MaAVOA in real-life problems. Since the optimal Pareto front is not known for real-life

**Table 4. Performance comparison between the proposed MaAVOA and other algorithms in terms of IGD.**

| PROB. | OBJ. | MAAVOA IGD | NSGA3 IGD | UNSGA3 IGD | MOEAD IGD | CTAEA IGD | AGEMOEA IGD |
|---|---|---|---|---|---|---|---|
| DTLZ1 | 3 | 1.54E-02(7.07E-07) | 1.55E-02 (3.18E-05) | 1.55E-02 (2.19E-05) | 1.56E-02 (1.52E-04) | 1.54E-02 (1.77E-05) | 3.68E-02 (6.76E-05) |
| | 5 | 5.28E-02 (8.27E-05) | 5.28E-02 (9.55E-05) | 5.28E-02 (2.30E-04) | 5.29E-02 (2.63E-04) | 5.31E-02 (7.78E-06) | 1.89E+00 (2.74E-05) |
| | 8 | 1.23E-01 (1.64E-03) | 1.22E-01 (2.69E-04) | 1.24E-01 (7.50E-04) | 1.21E-01 (4.74E-04) | 1.26E-01 (2.04E-03) | 2.53E+00 (5.22E-05) |
| | 10 | 1.41E-01 (8.70E-04) | 1.43E-01 (2.97E-04) | 1.43E-01 (1.09E-03) | 1.37E-01 (7.85E-04) | 1.42E-01 (1.13E-03) | 2.07E+00 (1.27E-05) |
| | 15 | 3.31E-01 (4.24E-02) | 2.18E-01 (3.54E-03) | 2.06E-01 (2.03E-02) | 1.35E-01 (2.40E-03) | 2.05E-01 (2.44E-02) | 6.95E-01 (1.03E-05) |
| DTLZ2 | 3 | 4.09E-02 (2.83E-06) | 4.09E-02 (7.07E-06) | 4.09E-02 (4.24E-06) | 4.09E-02 (7.07E-07) | 4.10E-02 (4.95E-06) | 5.31E-02 (4.17E-05) |
| | 5 | 1.65E-01 (1.34E-05) | 1.65E-01 (1.41E-05) | 1.65E-01 (4.24E-05) | 1.65E-01 (0.00E+00) | 1.65E-01 (6.36E-05) | 2.32E-01 (3.02E-05) |
| | 8 | 3.59E-01 (1.41E-05) | 3.60E-01 (4.67E-04) | 3.60E-01 (2.55E-04) | 3.59E-01 (1.41E-05) | 3.59E-01 (1.27E-04) | 4.25E-01 (7.25E-06) |
| | 10 | 4.59E-01 (2.62E-04) | 4.60E-01(1.98E-04) | 4.60E-01 (7.07E-05) | 4.58E-01 (2.12E-05) | 4.49E-01 (1.41E-05) | 6.13E-01 (8.46E-05) |
| | 15 | 6.34E-01 (8.49E-05) | 6.35E-01 (1.41E-05) | 6.35E-01 (1.20E-04) | 6.31E-01 (9.26E-04) | 6.32E-01 (9.33E-04) | 6.39E-01 (4.68E-05) |
| DTLZ3 | 3 | 4.10E-02 (1.40E-04) | 4.45E-02 (8.85E-04) | 4.31E-02 (1.47E-03) | 4.65E-02 (2.69E-03) | 5.21E-02 (3.65E-03) | 1.02E+00 (2.10E-05) |
| | 5 | 1.66E-01 (1.20E-03) | 1.66E-01(4.81E-04) | 1.84E-01 (2.44E-02) | 1.71E-01 (4.92E-03) | 2.81E+00 (4.74E-01) | 1.33E+01 (3.22E-06) |
| | 8 | 4.02E-01 (1.64E-02) | 4.34E-01 (2.84E-02) | 8.40E-01 (5.97E-01) | 7.46E-01 (5.39E-01) | 3.04E+00 (1.16E+00) | 1.37E+01 (8.85E-05) |
| | 10 | 4.60E-01 (1.04E-02) | 4.62E-01 (4.04E-03) | 4.64E-01 (4.04E-03) | 1.14E+00 (5.80E-03) | 1.50E+00 (3.65E-01) | 2.96E+01 (7.79E-05) |
| | 15 | 4.53E+00 (2.26E+00) | 6.51E-01 (1.70E-02) | 2.40E+00 (7.41E-01) | 1.29E+00 (6.36E-04) | 1.53E+00 (1.24E+00) | 2.07E+01 (1.77E-05) |
| DTLZ4 | 3 | 4.09E-02 (1.41E-06) | 4.09E-02 (5.66E-06) | 4.09E-02 (9.90E-06) | 5.41E-01 (0.00E+00) | 4.10E-02 (6.15E-05) | 5.16E-02 (7.16E-05) |
| | 5 | 1.65E-01 (1.41E-05) | 1.65E-01 (2.12E-05) | 1.65E-01 (1.41E-05) | 5.16E-01 (4.96E-01) | 1.65E-01 (9.19E-05) | 2.54E-01 (8.84E-06) |
| | 8 | 3.59E-01 (1.48E-04) | 3.59E-01 (3.54E-05) | 3.59E-01 (1.41E-05) | 4.11E-01 (7.41E-02) | 3.59E-01 (5.66E-05) | 4.52E-01 (3.35E-05) |
| | 10 | 4.58E-01 (4.24E-05) | 4.58E-01 (5.66E-05) | 4.58E-01 (7.07E-06) | 6.77E-01 (5.25E-02) | 4.57E-01 (0.00E+00) | 5.63E-01 (6.92E-05) |
| | 15 | 6.33E-01 (4.95E-05) | 6.32E-01 (7.07E-05) | 6.32E-01 (7.78E-05) | 7.31E-01 (1.93E-02) | 6.32E-01 (1.98E-04) | 5.79E-01 (8.86E-05) |
| DTLZ5 | 3 | 3.03E-02 (3.09E-03) | 2.83E-02 (7.62E-04) | 2.86E-02 (4.93E-04) | 2.54E-02 (4.95E-06) | 5.22E-03 (2.62E-04) | 6.71E-03 (3.80E-05) |
| | 5 | 1.35E-01 (2.50E-03) | 1.27E-01 (3.71E-02) | 1.50E-01 (3.56E-02) | 2.24E-02 (2.89E-04) | 6.67E-02 (4.29E-03) | 1.24E-01 (2.92E-05) |
| | 8 | 1.36E-01 (2.11E-03) | 3.87E-01 (3.41E-02) | 3.56E-01 (5.66E-03) | 6.87E-02 (2.12E-06) | 4.43E-01 (9.79E-03) | 1.85E-01 (1.50E-05) |
| | 10 | 1.77E-01 (5.42E-02) | 3.22E-01 (8.58E-02) | 3.76E-01 (8.99E-02) | 6.62E-02 (1.13E-05) | 4.06E-01 (5.14E-02) | 3.32E-01 (4.52E-05) |
| | 15 | 2.60E-01 (7.41E-02) | 3.45E-01 (1.66E-02) | 5.67E-01 (2.35E-02) | 1.41E-01 (1.56E-04) | 3.73E-01 (1.01E-01) | 5.24E-01 (2.14E-05) |
| DTLZ6 | 3 | 2.84E-02 (1.27E-03) | 2.93E-02 (6.02E-03) | 3.70E-02 (2.74E-03) | 2.54E-02 (1.27E-05) | 8.16E-02 (7.54E-02) | 5.70E-03 (3.10E-06) |
| | 5 | 7.54E-01 (1.25E-01) | 2.17E+00 (3.13E-01) | 1.92E+00 (1.76E-02) | 2.25E-02 (2.12E-06) | 3.82E+00 (2.40E-01) | 7.24E+00 (4.22E-05) |
| | 8 | 1.08E+00 (1.01E+00) | 3.28E+00 (6.02E-02) | 3.47E+00 (1.02E-01) | 6.87E-02 (2.26E-05) | 2.42E+00 (1.59E-01) | 7.50E+00 (2.72E-05) |
| | 10 | 1.15E+00 (2.20E-01) | 3.52E+00 (8.77E-02) | 3.57E+00 (2.98E-01) | 4.67E-01 (7.54E-01) | 2.25E+00 (1.30E-01) | 8.25E+00 (1.23E-05) |
| | 15 | 1.01E+00 (6.79E-03) | 3.51E+00 (6.75E-01) | 3.22E+00 (1.12E-02) | 1.40E-01 (3.54E-05) | 2.33E+00 (3.37E-02) | 7.01E+00 (4.99E-05) |
| DTLZ7 | 3 | 7.33E-02 (3.17E-04) | 7.28E-02 (4.70E-04) | 7.24E-02 (1.60E-04) | 1.02E-01 (7.50E-04) | 6.05E-02 (2.72E-04) | 4.29E-02 (5.31E-05) |
| | 5 | 3.73E-01 (1.17E-02) | 3.60E-01 (5.54E-03) | 3.59E-01 (1.98E-04) | 5.18E-01 (3.23E-02) | 2.86E-01 (4.07E-03) | 2.43E-01 (8.59E-05) |
| | 8 | 1.17E+00 (1.98E-03) | 1.27E+00 (1.23E-01) | 1.24E+00 (9.79E-02) | 2.30E+00 (2.39E-01) | 1.73E+00 (1.91E-02) | 6.06E-01 (3.00E-06) |
| | 10 | 1.94E+00 (4.24E-04) | 1.94E+00 (6.36E-03) | 1.98E+00 (6.10E-02) | 3.52E+00 (1.36E+00) | 3.35E+00 (9.16E-01) | 1.11E+00 (2.43E-05) |
| | 15 | 8.64E+00 (1.64E-02) | 8.37E+00 (5.82E-02) | 8.23E+00 (2.64E-01) | 4.23E+00 (1.49E+00) | 7.68E+00 (1.65E+00) | 2.43E+00 (8.12E-05) |
| +/ = /- | | | 10/3/22 | 11/3/21 | 17/3/15 | 6/3/26 | 9/2/24 |

applications, a reference set for the real-life problem was used for computing the IGD and HV, which was formed with the non-dominated solutions resulting from the union of all the approximation sets to the PF obtained by each algorithm at the end of every run.

## 5.1 Series-parallel system problem

The series-parallel system has subsystems in series and parallel combinations. Fig 6 shows an example of a series-parallel system with five subsystems where the final reliability function is divided into two parts. The first part contains subsystems 1 and 2, and the second part has subsystems 3, 4, and 5. For the first part, as subsystems are in series, therefore the product of $R_1$

**Table 5. Performance comparison between MaAVOA and other algorithms in terms of GD value on DTLZs.**

| | | MAAVOA | NSGA3 | UNSGA3 | MOEAD | CTAEA | AGEMOEA |
|---|---|---|---|---|---|---|---|
| PROB. | OBJ. | GD | GD | GD | GD | GD | GD |
| DTLZ1 | 3 | 1.48E-04 (5.87E-07) | 1.60E-04 (8.03E-06) | 1.56E-04 (4.57E-06) | 1.89E-04 (3.82E-05) | 1.51E-04 (3.51E-06) | 2.63E-02 (5.43E-05) |
| | 5 | 1.06E-03 (9.26E-06) | 1.11E-03 (8.09E-05) | 1.05E-03 (3.18E-06) | 1.06E-03 (1.07E-05) | 9.71E-04 (2.58E-05) | 7.27E+01 (5.94E-05) |
| | 8 | 3.05E-03 (2.76E-05) | 2.98E-03 (2.57E-05) | 3.26E-03 (2.47E-04) | 2.84E-03 (5.30E-05) | 3.31E-03 (2.27E-04) | 1.17E+02 (5.07E-05) |
| | 10 | 4.23E-04 (2.90E-04) | 1.43E-03 (1.06E-03) | 3.51E-04 (7.11E-05) | 9.50E-04 (1.22E-04) | 2.26E-03 (5.93E-05) | 1.42E+02 (2.81E-05) |
| | 15 | 1.59E-01 (1.94E-01) | 1.27E-03 (2.11E-04) | 1.31E-03 (2.41E-04) | 5.08E-03 (6.41E-04) | 5.17E-02 (6.26E-02) | 1.63E+02 (8.71E-05) |
| DTLZ2 | 3 | 3.97E-04 (1.34E-07) | 3.98E-04 (4.49E-06) | 3.95E-04 (2.40E-06) | 3.94E-04 (2.31E-06) | 3.94E-04 (7.25E-06) | 4.07E-02 (6.16E-05) |
| | 5 | 3.41E-03 (6.36E-07) | 3.40E-03 (8.49E-07) | 3.40E-03 (2.83E-06) | 3.41E-03 (2.62E-06) | 3.25E-03 (3.18E-06) | 2.01E-01 (3.15E-06) |
| | 8 | 9.64E-03 (2.83E-06) | 9.63E-03 (5.23E-06) | 9.62E-03 (4.52E-05) | 9.14E-03 (7.78E-07) | 7.62E-03 (5.29E-04) | 1.04E+00 (7.08E-05) |
| | 10 | 5.26E-04 (1.02E-04) | 1.09E-03 (2.31E-05) | 1.06E-03 (2.35E-05) | 5.58E-05 (1.58E-06) | 5.52E-03 (3.15E-04) | 1.22E+00 (7.78E-05) |
| | 15 | 1.08E-03 (6.25E-05) | 1.86E-03 (1.41E-04) | 2.08E-03 (2.23E-04) | 7.56E-04 (6.30E-04) | 9.81E-03 (6.26E-05) | 1.39E+00 (3.30E-05) |
| DTLZ3 | 3 | 4.32E-04 (3.40E-05) | 1.20E-03 (1.57E-05) | 2.31E-03 (1.56E-03) | 1.58E-03 (4.77E-04) | 2.17E-03 (4.40E-04) | 1.02E+00 (5.00E-05) |
| | 5 | 3.47E-03 (1.22E-04) | 3.53E-03 (1.53E-04) | 8.89E-03 (5.44E-03) | 4.12E-03 (7.42E-04) | 2.89E-01 (3.24E-02) | 1.34E+02 (3.30E-05) |
| | 8 | 2.65E-02 (1.75E-02) | 3.73E-02 (1.19E-02) | 5.91E-02 (5.97E-03) | 6.47E-03 (4.38E-03) | 4.76E-01 (2.10E-01) | 2.90E+02 (5.78E-05) |
| | 10 | 4.66E-03 (1.13E-03) | 5.88E-03 (2.86E-04) | 3.92E-03 (1.23E-03) | 3.19E-03 (1.36E-05) | 2.40E-01 (7.22E-02) | 3.94E+02 (3.26E-05) |
| | 15 | 9.93E-01 (5.21E-01) | 1.54E-02 (1.42E-02) | 2.60E-01 (8.41E-02) | 5.61E-04 (2.13E-04) | 6.92E-01 (9.10E-01) | 3.49E+02 (2.51E-05) |
| DTLZ4 | 3 | 3.95E-04 (2.83E-07) | 3.98E-04 (8.13E-07) | 3.95E-04 (2.26E-07) | 2.87E-04 (2.59E-05) | 3.95E-04 (3.73E-06) | 3.97E-02 (1.47E-05) |
| | 5 | 3.41E-03 (2.76E-06) | 3.40E-03 (6.22E-06) | 3.39E-03 (5.30E-06) | 2.69E-03 (9.75E-04) | 3.26E-03 (2.33E-05) | 3.63E-01 (2.12E-05) |
| | 8 | 9.65E-03 (1.19E-05) | 9.60E-03 (6.22E-06) | 9.58E-03 (3.25E-06) | 8.22E-03 (1.30E-03) | 9.05E-03 (1.02E-04) | 1.14E+00 (5.94E-05) |
| | 10 | 4.34E-03 (3.62E-03) | 2.82E-04 (9.56E-05) | 3.74E-04 (1.60E-05) | 1.82E-03 (6.69E-04) | 6.50E-04 (3.58E-05) | 1.50E+00 (5.47E-05) |
| | 15 | 8.78E-03 (6.95E-03) | 2.84E-04 (1.29E-04) | 2.38E-04 (5.08E-05) | 3.74E-03 (2.22E-03) | 1.01E-03 (4.58E-04) | 1.88E+00 (3.10E-06) |
| DTLZ5 | 3 | 1.41E-04 (9.89E-05) | 5.57E-04 (3.98E-04) | 3.55E-04 (9.25E-05) | 5.56E-06 (4.14E-08) | 1.24E-03 (1.60E-03) | 2.15E-03 (6.43E-05) |
| | 5 | 2.15E-01 (1.43E-02) | 1.97E-01 (9.21E-03) | 2.05E-01 (5.47E-03) | 2.50E-02 (1.97E-03) | 1.03E-01 (3.89E-04) | 2.05E+00 (7.52E-05) |
| | 8 | 1.93E-01 (2.11E-02) | 1.44E-01 (1.47E-03) | 1.56E-01 (4.34E-03) | 3.84E-03 (1.13E-05) | 1.22E-01 (2.43E-03) | 5.36E+00 (1.74E-05) |
| | 10 | 1.89E-01 (3.49E-02) | 1.75E-01 (2.75E-02) | 1.66E-01 (4.54E-02) | 2.06E-03 (3.56E-04) | 1.02E-01 (1.65E-03) | 9.10E+00 (4.80E-06) |
| | 15 | 1.67E-01 (8.93E-02) | 1.73E-01 (3.55E-02) | 1.97E-01 (3.26E-02) | 9.36E-07 (7.53E-08) | 1.30E-01 (7.00E-04) | 1.83E+01 (1.30E-05) |
| DTLZ6 | 3 | 9.91E-06 (1.72E-06) | 9.83E-06 (2.72E-08) | 3.07E-03 (4.33E-03) | 4.98E-06 (4.30E-07) | 2.03E-01 (2.72E-02) | 5.46E-03 (6.15E-05) |
| | 5 | 2.44E-01 (5.19E-02) | 2.69E-01 (3.52E-02) | 2.21E-01 (1.95E-02) | 2.84E-02 (6.20E-03) | 3.66E-01 (1.08E-02) | 1.92E+01 (1.03E-05) |
| | 8 | 2.73E-01 (6.31E-02) | 3.42E-01 (9.54E-03) | 3.46E-01 (1.82E-03) | 2.63E-06 (1.23E-07) | 2.98E-01 (9.92E-03) | 4.81E+01 (5.11E-05) |
| | 10 | 2.15E-01 (3.05E-03) | 2.65E-01 (3.34E-03) | 2.79E-01 (1.34E-03) | -5.00E-01 (7.07E-01) | 2.05E-01 (2.12E-02) | 8.11E+01 (4.96E-05) |
| | 15 | 5.20E-01 (8.22E-02) | 3.60E-01 (7.31E-02) | 3.40E-01 (4.99E-03) | 1.04E-06 (1.95E-07) | 2.71E-01 (1.42E-02) | 2.66E+02 (3.34E-05) |
| DTLZ7 | 3 | 5.49E-02 (7.42E-02) | 2.43E-03 (3.79E-04) | 2.09E-03 (2.10E-04) | 5.13E-03 (8.21E-04) | 2.34E-03 (1.34E-04) | 2.01E-02 (5.70E-05) |
| | 5 | 1.07E-01 (3.70E-02) | 1.42E-02 (1.81E-03) | 1.39E-02 (3.14E-04) | 3.53E-03 (1.49E-05) | 6.59E-03 (1.29E-03) | 1.40E-01 (2.25E-05) |
| | 8 | 7.00E-02 (7.73E-02) | 2.15E-01 (1.01E-01) | 1.60E-01 (1.47E-01) | 1.44E-02 (4.89E-04) | 2.07E-02 (1.10E-04) | 3.40E-01 (6.73E-05) |
| | 10 | 3.08E-02 (2.05E-02) | 2.00E-01 (1.04E-01) | 2.33E-01 (6.17E-02) | 1.59E-02 (3.75E-05) | 5.07E-02 (1.07E-02) | 8.69E-01 (1.78E-05) |
| | 15 | 6.87E-02 (3.74E-02) | 2.10E-01 (7.14E-02) | 1.77E-01 (5.87E-02) | 8.06E-02 (1.67E-02) | 3.18E-01 (1.75E-01) | 3.25E+00 (8.36E-05) |
| +/ = /- | | | 8/3/24 | 8/3/24 | 19/3/15 | 4/3/28 | 10/3/22 |

and $R_2$ is used. For the second part, $R_3$ and $R_4$ are parallel, so the function will be $R_3+R_4-R_3R_4$. The combination of $R_3$ and $R_4$ are in series with $R_5$. Therefore, the product of $(R_3+R_4-R_3R_4)$ and $R_5$ is used in the final function as shown in Eq (26). Volume and weight increase with extra components under permissible limits and restrictions. In Eq (27), $w_i$ represents the weight and $v_i$ represents the volume of component $i$ with $n$ number of redundant components. As shown in Eq (28), system cost $C_s$ also contains two additional factors $\alpha_i \left( -\frac{1000}{\log(r_i)} \right)^{\beta_i}$ and $exp$ (0.25$n_i$), where the first one represents the cost of a single component $i^{th}$ in the subsystem, and the second one is due to the cost of interconnecting hardware. In Eq (29), for system weight $W_s$, there is an extra factor

**Table 6. Performance comparison between MaAVOA and other algorithms in terms of HV value on DTLZs.**

| | | MAAVOA | NSGA3 | UNSGA3 | MOEAD | CTAEA | AGEMOEA |
|---|---|---|---|---|---|---|---|
| PROB. | OBJ. | HV | HV | HV | HV | HV | HV |
| DTLZ1 | 3 | 8.50E-01 (0.00E+00) | 8.49E-01 (4.10E-04) | 8.50E-01 (2.97E-04) | 8.48E-01 (1.34E-03) | 8.50E-01 (2.69E-04) | 8.42E-01 (1.45E-05) |
| | 5 | 9.79E-01 (2.12E-05) | 9.79E-01 (6.58E-04) | 9.79E-01 (5.59E-04) | 9.79E-01 (5.09E-04) | 9.79E-01 (2.47E-04) | 0.00E+00 (2.29E-05) |
| | 8 | 9.97E-01 (2.76E-04) | 9.97E-01 (8.49E-05) | 9.97E-01 (1.41E-05) | 9.97E-01 (8.49E-05) | 9.96E-01 (3.82E-04) | 0.00E+00 (2.59E-05) |
| | 10 | 1.00E+00 (2.12E-05) | 1.00E+00 (1.41E-05) | 1.00E+00 (4.95E-05) | 9.99E-01 (3.54E-05) | 1.00E+00 (7.07E-06) | 0.00E+00 (3.18E-05) |
| | 15 | 3.84E-01 (5.43E-01) | 9.99E-01 (2.05E-04) | 9.99E-01 (1.70E-04) | 8.95E-01 (1.52E-02) | 9.99E-01 (1.70E-04) | 5.56E-01 (3.76E-05) |
| DTLZ2 | 3 | 5.71E-01 (5.66E-05) | 5.71E-01 (7.07E-06) | 5.71E-01 (2.12E-05) | 5.71E-01 (1.41E-05) | 5.71E-01 (0.00E+00) | 5.55E-01 (1.14E-05) |
| | 5 | 8.12E-01 (4.10E-04) | 8.11E-01 (1.26E-03) | 8.12E-01 (3.96E-04) | 8.12E-01 (9.19E-05) | 8.12E-01 (1.13E-04) | 7.10E-01 (5.01E-05) |
| | 8 | 9.35E-01 (1.98E-04) | 9.33E-01 (9.97E-04) | 9.32E-01 (8.77E-04) | 9.36E-01 (3.75E-04) | 9.35E-01 (1.41E-04) | 7.22E-01 (3.72E-05) |
| | 10 | 9.74E-01 (3.32E-04) | 9.72E-01 (2.26E-04) | 9.72E-01 (1.91E-04) | 9.76E-01 (2.33E-04) | 9.74E-01 (9.90E-05) | 4.19E-01 (6.12E-05) |
| | 15 | 9.89E-01 (2.97E-04) | 9.88E-01 (4.10E-04) | 9.87E-01 (3.32E-04) | 9.90E-01 (2.05E-04) | 9.63E-01 (2.24E-03) | 6.03E-01 (8.50E-05) |
| DTLZ3 | 3 | 5.68E-01 (1.98E-03) | 5.53E-01 (6.36E-05) | 5.57E-01 (6.14E-03) | 5.47E-01 (9.22E-03) | 5.37E-01 (7.75E-03) | 0.00E+00 (2.13E-06) |
| | 5 | 8.03E-01 (1.09E-02) | 8.00E-01 (2.66E-03) | 7.70E-01 (3.32E-02) | 7.80E-01 (2.14E-02) | 0.00E+00 (0.00E+00) | 0.00E+00 (6.07E-05) |
| | 8 | 8.55E-01 (3.55E-02) | 7.79E-01 (5.99E-02) | 4.00E-01 (5.65E-01) | 5.10E-01 (5.76E-01) | 0.00E+00 (0.00E+00) | 0.00E+00 (2.57E-05) |
| | 10 | 9.63E-01 (1.16E-03) | 9.53E-01 (4.39E-03) | 9.60E-01 (5.69E-03) | 1.09E-01 (1.18E-03) | 1.51E-03 (2.13E-03) | 0.00E+00 (7.48E-05) |
| | 15 | 0.00E+00 (0.00E+00) | 9.48E-01 (4.23E-02) | 0.00E+00 (0.00E+00) | 8.75E-02 (3.67E-04) | 4.44E-01 (6.28E-01) | 0.00E+00 (8.76E-05) |
| DTLZ4 | 3 | 5.71E-01 (7.07E-06) | 5.71E-01 (1.41E-04) | 5.71E-01 (0.00E+00) | 3.45E-01 (4.95E-05) | 5.71E-01 (2.12E-05) | 5.55E-01 (2.65E-05) |
| | 5 | 8.12E-01 (6.36E-04) | 8.12E-01 (1.20E-04) | 8.12E-01 (5.73E-04) | 5.69E-01 (3.45E-01) | 8.12E-01 (1.27E-04) | 7.02E-01 (8.29E-05) |
| | 8 | 9.36E-01 (1.91E-04) | 9.35E-01 (2.83E-05) | 9.35E-01 (4.81E-04) | 9.17E-01 (2.64E-02) | 9.36E-01 (1.98E-04) | 7.12E-01 (4.98E-05) |
| | 10 | 9.75E-01 (5.66E-05) | 9.75E-01 (4.24E-05) | 9.75E-01 (1.27E-04) | 8.71E-01 (3.99E-02) | 9.75E-01 (4.03E-04) | 7.25E-01 (5.04E-05) |
| | 15 | 9.90E-01 (7.78E-05) | 9.90E-01 (1.13E-04) | 9.90E-01 (1.34E-04) | 9.48E-01 (1.30E-02) | 9.90E-01 (9.19E-05) | 8.18E-01 (1.62E-05) |
| DTLZ5 | 3 | 1.84E-01 (1.59E-03) | 1.86E-01 (2.33E-04) | 1.86E-01 (3.46E-04) | 1.87E-01 (1.41E-05) | 1.99E-01 (2.12E-05) | 1.98E-01 (6.66E-06) |
| | 5 | 6.10E-02 (6.16E-03) | 5.13E-02 (1.09E-03) | 5.28E-02 (3.40E-02) | 1.27E-01 (4.31E-04) | 1.13E-01 (7.87E-03) | 1.11E-01 (1.23E-05) |
| | 8 | 2.61E-02 (2.92E-02) | 1.47E-05 (2.08E-05) | 4.62E-04 (6.53E-04) | 1.04E-01 (1.13E-04) | 9.08E-03 (1.28E-02) | 9.67E-02 (1.12E-05) |
| | 10 | 1.61E-02 (2.19E-02) | 1.54E-05 (1.93E-05) | 2.04E-05 (2.89E-05) | 9.96E-02 (6.08E-05) | 1.19E-04 (1.68E-04) | 9.19E-02 (7.83E-05) |
| | 15 | 9.10E-02 (8.10E-04) | 2.40E-02 (3.40E-02) | 0.00E+00 (0.00E+00) | 9.43E-02 (2.72E-04) | 5.61E-02 (2.95E-03) | 9.12E-02 (8.88E-05) |
| DTLZ6 | 3 | 1.86E-01 (1.13E-04) | 1.85E-01 (1.59E-03) | 1.84E-01 (1.35E-03) | 1.87E-01 (7.07E-06) | 1.30E-01 (5.86E-02) | 2.00E-01 (4.32E-05) |
| | 5 | 0.00E+00 (0.00E+00) | 0.00E+00 (0.00E+00) | 0.00E+00 (0.00E+00) | 1.27E-01 (4.45E-04) | 0.00E+00 (0.00E+00) | 0.00E+00 (3.08E-05) |
| | 8 | 4.99E-04 (7.06E-04) | 0.00E+00 (0.00E+00) | 0.00E+00 (0.00E+00) | 1.04E-01 (2.33E-04) | 0.00E+00 (0.00E+00) | 0.00E+00 (3.69E-05) |
| | 10 | 0.00E+00 (0.00E+00) | 0.00E+00 (0.00E+00) | 0.00E+00 (0.00E+00) | 4.50E-01 (7.78E-01) | 0.00E+00 (0.00E+00) | 0.00E+00 (5.81E-06) |
| | 15 | 0.00E+00 (0.00E+00) | 0.00E+00 (0.00E+00) | 0.00E+00 (0.00E+00) | 9.44E-02 (2.33E-04) | 0.00E+00 (0.00E+00) | 0.00E+00 (5.23E-05) |
| DTLZ7 | 3 | 2.76E-01 (6.36E-05) | 2.76E-01 (1.34E-04) | 2.76E-01 (3.89E-04) | 2.63E-01 (2.83E-04) | 2.78E-01 (3.61E-04) | 2.76E-01 (7.61E-05) |
| | 5 | 2.50E-01 (3.87E-03) | 2.45E-01 (3.82E-03) | 2.46E-01 (4.84E-03) | 1.42E-01 (1.17E-03) | 2.15E-01 (3.44E-03) | 1.97E-01 (7.07E-05) |
| | 8 | 1.83E-01 (6.51E-04) | 1.22E-01 (3.69E-02) | 1.54E-01 (2.19E-03) | 5.78E-04 (2.77E-04) | 7.74E-02 (1.16E-02) | 3.17E-02 (5.06E-05) |
| | 10 | 1.75E-01 (4.45E-04) | 1.52E-01 (5.59E-04) | 1.26E-01 (1.07E-02) | 1.12E-04 (1.47E-04) | 3.38E-02 (4.79E-02) | 3.73E-03 (7.89E-05) |
| | 15 | 1.49E-01 (3.32E-03) | 9.76E-02 (1.23E-02) | 1.04E-01 (2.17E-02) | 2.98E-06 (4.21E-06) | 1.75E-10 (2.21E-10) | 4.55E-12 (8.73E-06) |
| +/ = /- | | | 8/4/24 | 9/5/22 | 17/0/18 | 11/5/19 | 23/5/7 |

$exp$ $(0.25n_i)$ for the interconnecting hardware. The mathematical formulation of the problem is a nonlinear mixed-integer programming problem given as follows:

$$\max f(r, n) = 1 - (1 - R_1 R_2)(1 - (R_3 + R_4 - R_3 R_4) \tag{26}$$

$$\min V_s(n) = \sum_{i=m}^{m} w_i v_i^2 n_i^2 \tag{27}$$

$$\min C_s(r, n) = \sum_{i=m}^{m} \alpha_i \left( -\frac{1000}{\log(r_i)} \right)^{\beta_i} [n_i + \exp(0.25n_i)] \tag{28}$$

**Table 7. The performance metrics comparison between MaAVOA and other algorithms in terms of IGD value on DTLZs in the case of the 100000 function evaluations.**

| | | MAAVOA | NSGA3 | UNSGA3 | MOEAD | CTAEA | AGEMOEA |
|---|---|---|---|---|---|---|---|
| PROB. | OBJ | IGD | IGD | IGD | IGD | IGD | IGD |
| DTLZ1 | 3 | 1.54E-02 (4.24E-06) | 1.54E-02 (3.54E-06) | 1.54E-02 (1.70E-05) | 1.54E-02 (1.27E-05) | 1.54E-02 (3.54E-06) | 3.67E-02 (6.90E-05) |
| | 5 | 5.32E-02 (3.74E-04) | 5.27E-02 (2.26E-05) | 5.29E-02 (3.17E-04) | 5.30E-02 (1.20E-04) | 5.32E-02 (1.20E-04) | 2.67E-01 (3.24E-05) |
| | 8 | 1.22E-01 (4.74E-04) | 1.21E-01 (4.24E-05) | 1.23E-01 (1.91E-04) | 1.21E-01 (7.07E-06) | 1.21E-01 (3.61E-04) | 4.96E-01 (7.53E-05) |
| | 10 | 2.21E-01 (2.69E-03) | 1.42E-01 (5.37E-04) | 1.43E-01 (4.10E-04) | 1.29E-01 (7.85E-04) | 1.43E-01 (1.46E-03) | 6.84E-01 (2.56E-05) |
| | 15 | 7.38E-01 (4.16E-01) | 2.24E-01 (3.59E-03) | 2.16E-01 (9.57E-03) | 1.56E-01 (6.74E-03) | 1.99E-01 (3.30E-02) | 4.80E-01 (4.75E-05) |
| DTLZ2 | 3 | 4.09E-02 (0.00E+0) | 4.09E-02 (2.26E-05) | 4.09E-02 (4.95E-06) | 4.09E-02 (7.07E-07) | 4.09E-02 (3.54E-06) | 5.37E-02 (5.82E-05) |
| | 5 | 1.65E-01 (2.12E-05) | 1.65E-01 (7.07E-06) | 1.65E-01 (2.12E-05) | 1.65E-01 (7.07E-06) | 1.65E-01 (4.24E-05) | 2.49E-01 (1.57E-05) |
| | 8 | 3.59E-01 (7.78E-05) | 3.59E-01 (8.49E-05) | 3.59E-01 (3.54E-05) | 3.59E-01 (0.00E+0) | 3.58E-01 (1.56E-04) | 3.97E-01 (3.86E-06) |
| | 10 | 4.61E-01 (2.40E-04) | 4.61E-01 (5.37E-04) | 4.61E-01 (4.95E-05) | 4.58E-01 (8.49E-05) | 4.50E-01 (1.57E-03) | 5.29E-01 (3.89E-05) |
| | 15 | 6.34E-01 (2.05E-04) | 6.34E-01 (7.07E-06) | 6.34E-01 (1.27E-04) | 6.32E-01 (2.83E-05) | 6.30E-01 (1.43E-03) | 5.64E-01 (3.93E-05) |
| DTLZ3 | 3 | 4.10E-02 (4.24E-06) | 4.14E-02 (2.80E-04) | 4.12E-02 (5.30E-05) | 4.13E-02 (3.54E-04) | 4.24E-02 (6.94E-04) | 4.99E-02 (5.39E-05) |
| | 5 | 1.67E-01 (3.63E-03) | 1.93E-01 (3.82E-02) | 1.67E-01 (8.63E-04) | 1.69E-01 (3.27E-03) | 2.96E+0 (2.76E-01) | 5.22E+00 (2.26E-05) |
| | 8 | 1.99E+0 (1.76E+0) | 4.25E-01 (7.32E-02) | 9.62E-01 (2.87E-01) | 3.61E-01 (1.13E-04) | 2.51E+0 (1.85E+0) | 1.19E+01 (8.72E-05) |
| | 10 | 3.88E+0 (2.83E+0) | 5.44E-01 (8.77E-02) | 5.06E-01 (3.45E-02) | 1.16E+0 (4.24E-04) | 3.24E+0 (9.70E-01) | 1.80E+01 (5.49E-05) |
| | 15 | 6.26E+0 (4.99E+0) | 6.51E-01 (4.12E-03) | 1.03E+0 (5.35E-01) | 1.27E+0 (2.43E-02) | 6.40E-01 (4.70E-03) | 5.58E+00 (4.74E-05) |
| DTLZ4 | 3 | 4.09E-02 (0.00E+0) | 4.09E-02 (3.54E-06) | 4.09E-02 (3.54E-06) | 2.91E-01 (3.54E-01) | 4.10E-02 (8.49E-06) | 5.40E-02 (2.31E-05) |
| | 5 | 1.65E-01 (2.12E-05) | 1.65E-01 (7.07E-06) | 1.65E-01 (2.12E-05) | 6.37E-01 (3.25E-01) | 1.65E-01 (1.70E-04) | 2.62E-01 (4.26E-05) |
| | 8 | 3.59E-01 (6.08E-04) | 3.59E-01 (7.78E-05) | 3.59E-01 (6.36E-05) | 5.16E-01 (2.22E-01) | 3.59E-01 (9.19E-05) | 4.37E-01 (4.19E-05) |
| | 10 | 4.58E-01 (0.00E+0) | 4.57E-01 (3.11E-04) | 4.57E-01 (3.54E-05) | 7.14E-01 (2.05E-04) | 4.56E-01 (9.19E-05) | 5.31E-01 (7.95E-05) |
| | 15 | 6.33E-01 (9.19E-05) | 6.32E-01 (7.07E-06) | 6.32E-01 (7.07E-06) | 7.80E-01 (1.28E-01) | 6.32E-01 (1.34E-04) | 5.68E-01 (7.30E-05) |
| DTLZ5 | 3 | 2.97E-02 (2.11E-03) | 2.94E-02 (1.92E-04) | 2.90E-02 (4.62E-04) | 2.54E-02 (7.07E-07) | 5.33E-03 (9.98E-05) | 6.79E-03 (2.11E-05) |
| | 5 | 1.18E-01 (7.38E-03) | 1.41E-01 (2.22E-02) | 1.90E-01 (6.70E-02) | 2.25E-02 (1.16E-04) | 7.23E-02 (1.33E-02) | 1.34E-01 (6.16E-05) |
| | 8 | 1.24E-01 (1.56E-04) | 3.51E-01 (5.66E-02) | 2.91E-01 (3.10E-02) | 6.87E-02 (8.49E-06) | 4.31E-01 (1.27E-01) | 2.58E-01 (3.40E-05) |
| | 10 | 1.46E-01 (2.64E-02) | 3.98E-01 (2.21E-02) | 3.84E-01 (8.56E-02) | 6.62E-02 (3.46E-05) | 3.96E-01 (4.37E-02) | 2.86E-01 (3.32E-05) |
| | 15 | 3.55E-01 (2.18E-01) | 3.70E-01 (1.03E-01) | 4.35E-01 (2.64E-02) | 1.41E-01 (2.83E-05) | 4.09E-01 (2.47E-03) | 3.87E-01 (7.44E-05) |
| DTLZ6 | 3 | 2.58E-02 (1.04E-03) | 2.80E-02 (8.90E-04) | 3.21E-02 (4.80E-03) | 2.55E-02 (2.12E-06) | 2.60E-02 (2.10E-04) | 6.33E-03 (7.90E-05) |
| | 5 | 1.51E+0 (1.51E-02) | 2.09E+0 (8.95E-02) | 2.01E+0 (3.20E-02) | 2.25E-02 (2.83E-06) | 3.96E+0 (1.34E-01) | 4.45E+00 (6.65E-06) |
| | 8 | 1.83E+0 (1.00E-01) | 2.63E+0 (2.80E-01) | 2.74E+0 (2.15E-02) | 6.86E-02 (2.40E-05) | 1.75E+0 (1.94E-01) | 6.71E+00 (8.17E-05) |
| | 10 | 3.37E+0 (6.65E-01) | 4.86E+0 (4.93E-02) | 4.98E+0 (1.68E-02) | 6.62E-02 (9.90E-06) | 3.80E+0 (2.79E-01) | 6.91E+00 (8.40E-05) |
| | 15 | 7.36E-01 (1.52E-02) | 1.54E+0 (1.41E-01) | 1.59E+0 (2.20E-01) | 1.40E-01 (1.20E-04) | 1.07E+0 (9.08E-02) | 6.41E+00 (6.74E-05) |
| DTLZ7 | 3 | 7.32E-02 (6.20E-04) | 7.21E-02 (6.83E-04) | 7.23E-02 (6.60E-04) | 1.02E-01 (2.76E-04) | 6.30E-02 (2.41E-03) | 4.34E-02 (1.57E-05) |
| | 5 | 3.71E-01 (1.83E-02) | 3.61E-01 (2.55E-04) | 3.56E-01 (1.46E-03) | 4.94E-01 (8.06E-04) | 2.80E-01 (9.72E-03) | 2.27E-01 (8.42E-05) |
| | 8 | 1.17E+0 (6.36E-04) | 1.28E+0 (3.88E-02) | 1.27E+0 (1.33E-01) | 2.84E+0 (9.98E-01) | 1.58E+0 (2.92E-01) | 5.53E-01 (6.65E-05) |
| | 10 | 2.00E+0 (9.33E-03) | 4.78E-01 (2.09E+0) | 1.93E+0 (5.66E-03) | 2.99E+0 (6.27E-01) | 3.73E+0 (1.49E+0) | 9.57E-01 (3.62E-05) |
| | 15 | 8.62E+0 (3.91E-02) | 8.37E+0 (2.64E-02) | 8.47E+0 (3.36E-02) | 3.44E+0 (4.02E-01) | 8.83E+0 (1.07E-01) | 2.12E+00 (8.98E-05) |
| +/ = /- | | | 18/3/14 | 18/3/14 | 12/3/20 | 23/3/9 | 11/0/24 |

$$\min W_s(n) = w_i n_i \exp(0.25 n_i) \qquad (29)$$

$$s.t. V_s - V \le 0, C_s - C \le 0, W_s - W \le 0 \text{ with } 0 \le r_i \ge 1, \quad n_i \in \mathbb{Z}^+, 1 \le i \ge m$$

where $R_i(n_i) = 1 - (1 - r_i)^{n_i}$ for the $i^{th}$ subsystem, $\alpha_i$ and $\beta_i$ are constraints representing the physical characteristic of each component at stage $i$.

**Table 8. The performance metrics comparison between MaAVOA and other algorithms in terms of GD value on DTLZs in the case of the 100000 function evaluations.**

| PROB. | OBJ. | MAAVOA GD | NSGA3 GD | UNSGA3 GD | MOEAD GD | CTAEA GD | AGEMOEA GD |
|---|---|---|---|---|---|---|---|
| DTLZ1 | 3 | 1.48E-04 (4.60E-07) | 1.50E-04 (6.01E-07) | 8.49E-03 (1.18E-02) | 1.49E-04 (2.02E-06) | 1.48E-04 (1.90E-06) | 2.52E-02 (6.62E-06) |
| | 5 | 1.04E-03 (9.62E-06) | 1.11E-03 (6.95E-05) | 1.05E-03 (2.70E-05) | 1.06E-03 (7.42E-06) | 9.42E-04 (6.21E-06) | 3.49E+01 (4.20E-05) |
| | 8 | 2.99E-03 (2.12E-07) | 2.97E-03 (3.39E-06) | 3.62E-03 (8.65E-04) | 2.87E-03 (4.12E-05) | 2.48E-03 (5.69E-05) | 9.04E+01 (8.53E-05) |
| | 10 | 1.93E-02 (2.01E-02) | 4.40E-04 (3.09E-05) | 1.23E-03 (5.56E-04) | 1.52E-03 (1.09E-04) | 2.70E-03 (2.60E-04) | 1.35E+02 (1.70E-05) |
| | 15 | 1.87E-01 (1.24E-01) | 6.88E-04 (4.26E-04) | 1.07E-03 (7.70E-04) | 4.27E-03 (5.64E-04) | 4.00E-02 (4.13E-03) | 1.36E+02 (7.86E-05) |
| DTLZ2 | 3 | 3.96E-04 (3.53E-06) | 4.11E-04 (2.05E-05) | 3.98E-04 (4.52E-06) | 3.95E-04 (6.01E-07) | 3.97E-04 (3.73E-06) | 4.04E-02 (1.16E-05) |
| | 5 | 3.40E-03 (4.67E-06) | 3.40E-03 (4.95E-07) | 3.40E-03 (4.74E-06) | 3.41E-03 (2.55E-06) | 3.25E-03 (4.17E-05) | 2.88E-01 (4.83E-05) |
| | 8 | 9.64E-03 (8.13E-06) | 9.63E-03 (1.87E-05) | 9.62E-03 (6.58E-06) | 9.15E-03 (3.54E-06) | 8.82E-03 (2.79E-05) | 9.02E-01 (6.16E-05) |
| | 10 | 1.46E-03 (1.40E-04) | 1.78E-03 (1.31E-04) | 1.64E-03 (1.19E-04) | 1.39E-04 (5.73E-07) | 6.77E-03 (2.65E-04) | 1.25E+00 (5.42E-05) |
| | 15 | 1.12E-03 (2.46E-04) | 1.01E-03 (3.13E-05) | 1.30E-03 (3.02E-04) | 9.63E-05 (1.30E-06) | 1.01E-02 (3.46E-04) | 1.39E+00 (4.15E-05) |
| DTLZ3 | 3 | 4.09E-04 (5.69E-06) | 5.03E-04 (1.04E-04) | 4.83E-04 (1.67E-05) | 5.25E-04 (1.30E-04) | 5.67E-04 (1.11E-04) | 3.78E-02 (5.89E-05) |
| | 5 | 3.67E-03 (3.90E-04) | 1.05E-02 (9.73E-03) | 3.44E-03 (1.74E-04) | 3.96E-03 (4.89E-04) | 3.24E-01 (2.47E-02) | 6.21E+01 (5.93E-05) |
| | 8 | 2.24E-01 (1.43E-01) | 4.17E-02 (4.28E-02) | 8.81E-02 (9.64E-04) | 9.24E-03 (2.06E-05) | 3.29E-01 (2.16E-01) | 2.56E+02 (5.27E-05) |
| | 10 | 4.52E-01 (2.17E-01) | 3.56E-02 (3.80E-02) | 1.75E-02 (7.18E-03) | 2.69E-03 (7.07E-07) | 6.60E-01 (5.83E-02) | 4.41E+02 (1.53E-05) |
| | 15 | 1.14E+0 (1.02E+0) | 1.90E-02 (1.71E-02) | 6.64E-02 (5.20E-02) | 1.17E-03 (1.09E-03) | 5.73E-02 (1.25E-02) | 3.40E+02 (1.97E-05) |
| DTLZ4 | 3 | 3.97E-04 (1.62E-06) | 3.97E-04 (2.92E-06) | 3.96E-04 (2.58E-06) | 3.45E-04 (7.11E-05) | 3.92E-04 (1.70E-06) | 3.98E-02 (4.55E-05) |
| | 5 | 3.39E-03 (1.31E-05) | 3.40E-03 (1.41E-07) | 3.39E-03 (5.66E-06) | 2.40E-03 (7.42E-04) | 3.23E-03 (1.11E-05) | 3.43E-01 (6.73E-05) |
| | 8 | 1.05E-02 (1.23E-03) | 9.63E-03 (9.69E-06) | 9.62E-03 (7.00E-06) | 7.24E-03 (2.69E-03) | 9.15E-03 (5.52E-05) | 1.10E+00 (1.81E-05) |
| | 10 | 7.15E-03 (1.29E-03) | 4.50E-04 (1.73E-05) | 4.88E-04 (2.31E-05) | 2.36E-03 (9.53E-04) | 1.17E-03 (4.67E-05) | 1.37E+00 (6.61E-05) |
| | 15 | 5.02E-03 (2.49E-03) | 3.00E-04 (1.08E-04) | 1.58E-04 (1.65E-05) | 3.10E-03 (1.27E-03) | 3.15E-04 (1.63E-04) | 1.80E+00 (2.11E-06) |
| DTLZ5 | 3 | 6.83E-04 (8.02E-04) | 1.26E-03 (1.42E-04) | 4.66E-04 (6.52E-05) | 5.56E-06 (6.26E-08) | 2.36E-04 (1.98E-05) | 2.61E-03 (2.46E-05) |
| | 5 | 2.03E-01 (8.06E-04) | 2.21E-01 (2.17E-02) | 2.09E-01 (4.29E-03) | 2.18E-02 (2.70E-03) | 1.04E-01 (1.29E-03) | 2.00E+00 (8.54E-05) |
| | 8 | 1.96E-01 (2.96E-02) | 1.55E-01 (7.88E-03) | 1.52E-01 (9.40E-03) | 3.73E-03 (4.13E-04) | 1.24E-01 (2.04E-03) | 5.00E+00 (5.03E-05) |
| | 10 | 1.75E-01 (3.62E-03) | 1.59E-01 (1.07E-02) | 1.80E-01 (3.22E-02) | 1.22E-03 (2.75E-04) | 9.81E-02 (1.05E-03) | 8.82E+00 (8.37E-05) |
| | 15 | 1.62E-01 (8.98E-02) | 1.90E-01 (5.05E-02) | 1.93E-01 (2.46E-03) | 9.04E-07 (7.78E-08) | 1.33E-01 (4.18E-03) | 2.71E+01 (3.01E-05) |
| DTLZ6 | 3 | 1.04E-05 (1.36E-06) | 1.03E-05 (1.05E-06) | 9.97E-06 (8.00E-07) | 5.39E-06 (1.45E-07) | 1.96E-01 (8.44E-02) | 2.15E-02 (7.69E-05) |
| | 5 | 1.94E-01 (5.72E-03) | 2.50E-01 (3.32E-04) | 2.43E-01 (2.14E-02) | 3.29E-02 (3.77E-04) | 3.72E-01 (9.11E-03) | 1.30E+01 (8.74E-05) |
| | 8 | 2.89E-01 (5.09E-02) | 2.76E-01 (1.23E-02) | 2.79E-01 (1.09E-02) | 2.49E-06 (5.44E-07) | 2.31E-01 (2.07E-02) | 4.52E+01 (4.85E-05) |
| | 10 | 2.67E-01 (1.74E-02) | 3.53E-01 (6.70E-03) | 3.60E-01 (2.26E-04) | 1.91E-06 (4.63E-08) | 3.00E-01 (1.40E-02) | 7.28E+01 (8.10E-05) |
| | 15 | 6.01E-01 (2.62E-02) | 2.23E-01 (1.76E-02) | 2.76E-01 (8.84E-02) | 7.23E-07 (1.40E-07) | 2.12E-01 (7.59E-03) | 2.31E+02 (7.15E-05) |
| DTLZ7 | 3 | 3.89E-03 (2.54E-03) | 2.08E-03 (5.29E-05) | 2.28E-03 (3.59E-04) | 4.45E-03 (1.72E-03) | 2.37E-03 (4.63E-05) | 2.03E-02 (1.47E-05) |
| | 5 | 6.78E-02 (4.57E-02) | 1.31E-02 (4.63E-04) | 1.40E-02 (1.01E-03) | 3.54E-03 (1.23E-05) | 6.57E-03 (1.81E-03) | 1.25E-01 (5.72E-05) |
| | 8 | 5.03E-02 (3.03E-02) | 1.71E-01 (1.18E-01) | 1.95E-01 (1.28E-02) | 1.46E-02 (3.90E-04) | 2.25E-02 (3.10E-03) | 2.36E-01 (3.71E-05) |
| | 10 | 1.53E-02 (2.35E-04) | -3.99E-01 (8.50E-01) | 1.77E-01 (1.17E-02) | 1.59E-02 (1.56E-04) | 6.53E-02 (1.19E-02) | 5.56E-01 (7.95E-05) |
| | 15 | 1.22E-01 (4.68E-02) | 2.08E-01 (1.76E-02) | 3.00E-01 (4.20E-02) | 9.43E-02 (2.17E-03) | 3.28E-01 (3.04E-04) | 1.84E+00 (3.64E-06) |
| +/ = /- | | | 19/3/13 | 19/3/13 | 9/3/22 | 25/3/7 | 0/0/35 |

Table 14 provides the input data for a series-parallel system where $r_i$, $\alpha_i$ and $\beta_i$ are uniformly generated from the ranges [0.95,1.0], [6,10], [1,5], and [11,20] respectively.

The algorithms are terminated after 250, 500, 1000, 2000, 4000, and 5000 generations. The engineering problem has 4 objective functions. Accordingly, the population size is chosen to be 969 ($Nr_1 = 16$, $Nr_2 = 0$, and $nRef = 969$).

In Table 15, the values of the performance measures for a series-parallel system with five subsystems are presented. NSGA-III and U-NSGA-III have performed better in terms of GD and IGD. In terms of HV, MaAVOA is better. CTAEA and AGEMOEA have the worst performance in all metrics. Fig 7 shows the final solution set obtained for all algorithms.

**Table 9. The performance metrics comparison between MaAVOA and other algorithms in terms of HV value on DTLZs in the case of the 100000 function evaluations.**

| | | MAAVOA | NSGA3 | UNSGA3 | MOEAD | CTAEA | AGEMOEA |
|---|---|---|---|---|---|---|---|
| PROB. | OBJ. | HV | HV | HV | HV | HV | HV |
| DTLZ1 | 3 | 8.50E-01 (8.49E-05) | 8.50E-01 (6.36E-05) | 8.50E-01 (1.70E-04) | 8.50E-01 (2.19E-04) | 8.50E-01 (3.54E-05) | 8.42E-01 (4.91E-05) |
| | 5 | 9.79E-01 (4.67E-04) | 9.79E-01 (2.05E-04) | 9.79E-01 (3.46E-04) | 9.79E-01 (1.34E-04) | 9.79E-01 (0.00E+0) | 8.28E-01 (4.32E-05) |
| | 8 | 9.97E-01 (1.41E-05) | 9.97E-01 (7.07E-06) | 9.97E-01 (2.76E-04) | 9.97E-01 (8.49E-05) | 9.97E-01 (0.00E+0) | 6.88E-01 (8.07E-05) |
| | 10 | 9.20E-01 (5.58E-02) | 9.99E-01 (7.07E-06) | 9.99E-01 (2.12E-05) | 9.98E-01 (2.19E-04) | 9.99E-01 (5.66E-05) | 2.70E-01 (6.31E-05) |
| | 15 | 1.47E-01 (2.08E-01) | 1.00E+0 (2.12E-05) | 1.00E+0 (1.27E-04) | 9.81E-01 (6.33E-03) | 9.99E-01 (5.23E-04) | 9.14E-01 (1.54E-05) |
| DTLZ2 | 3 | 5.71E-01 (1.41E-05) | 5.71E-01 (9.90E-05) | 5.71E-01 (2.12E-05) | 5.71E-01 (0.00E+0) | 5.71E-01 (0.00E+0) | 5.55E-01 (6.94E-05) |
| | 5 | 8.12E-01 (7.71E-04) | 8.11E-01 (3.89E-04) | 8.11E-01 (4.95E-05) | 8.12E-01 (7.78E-05) | 8.12E-01 (9.90E-04) | 7.07E-01 (3.35E-05) |
| | 8 | 9.34E-01 (2.33E-04) | 9.34E-01 (7.78E-05) | 9.34E-01 (3.04E-04) | 9.36E-01 (1.41E-04) | 9.36E-01 (7.00E-04) | 7.91E-01 (2.11E-05) |
| | 10 | 9.71E-01 (4.67E-04) | 9.70E-01 (9.90E-05) | 9.70E-01 (2.62E-04) | 9.75E-01 (2.83E-04) | 9.69E-01 (1.92E-03) | 8.02E-01 (7.97E-05) |
| | 15 | 9.89E-01 (3.25E-04) | 9.89E-01 (2.33E-04) | 9.89E-01 (4.17E-04) | 9.90E-01 (1.41E-04) | 9.71E-01 (3.01E-03) | 8.39E-01 (8.74E-05) |
| DTLZ3 | 3 | 5.69E-01 (6.36E-05) | 5.66E-01 (2.69E-03) | 5.67E-01 (3.54E-04) | 5.66E-01 (3.06E-03) | 5.64E-01 (2.80E-03) | 5.57E-01 (1.33E-05) |
| | 5 | 7.96E-01 (1.76E-02) | 7.62E-01 (5.08E-02) | 7.95E-01 (4.84E-03) | 7.85E-01 (1.49E-02) | 0.00E+0 (0.00E+0) | 0.00E+00 (1.19E-05) |
| | 8 | 1.66E-01 (2.34E-01) | 8.07E-01 (1.20E-01) | 1.64E-01 (2.31E-01) | 9.28E-01 (3.82E-04) | 2.01E-02 (2.85E-02) | 0.00E+00 (8.83E-05) |
| | 10 | 0.00E+0 (0.00E+0) | 7.82E-01 (1.99E-01) | 8.43E-01 (7.70E-02) | 1.03E-01 (1.74E-03) | 0.00E+0 (0.00E+0) | 0.00E+00 (2.14E-05) |
| | 15 | 0.00E+0 (0.00E+0) | 9.56E-01 (2.08E-02) | 4.71E-01 (6.66E-01) | 9.31E-02 (3.20E-03) | 9.51E-01 (8.49E-03) | 0.00E+00 (6.07E-05) |
| DTLZ4 | 3 | 5.71E-01 (3.54E-05) | 5.71E-01 (5.66E-05) | 5.71E-01 (4.95E-05) | 4.58E-01 (1.60E-01) | 5.71E-01 (2.83E-05) | 5.55E-01 (3.02E-05) |
| | 5 | 8.12E-01 (2.90E-04) | 8.12E-01 (4.10E-04) | 8.12E-01 (3.54E-04) | 5.18E-01 (2.77E-01) | 8.12E-01 (4.74E-04) | 7.07E-01 (5.42E-05) |
| | 8 | 9.36E-01 (2.69E-04) | 9.35E-01 (2.33E-04) | 9.36E-01 (3.68E-04) | 8.47E-01 (1.25E-01) | 9.36E-01 (4.88E-04) | 7.41E-01 (6.44E-05) |
| | 10 | 9.75E-01 (7.78E-05) | 9.75E-01 (4.10E-04) | 9.75E-01 (1.84E-04) | 8.43E-01 (3.75E-04) | 9.75E-01 (2.69E-04) | 7.88E-01 (5.58E-05) |
| | 15 | 9.90E-01 (6.36E-05) | 9.90E-01 (1.41E-05) | 9.91E-01 (5.66E-05) | 8.98E-01 (1.03E-01) | 9.90E-01 (1.48E-04) | 8.63E-01 (2.48E-05) |
| DTLZ5 | 3 | 1.83E-01 (1.41E-03) | 1.84E-01 (1.41E-05) | 1.85E-01 (2.26E-04) | 1.87E-01 (0.00E+0) | 1.99E-01 (7.07E-05) | 1.98E-01 (4.42E-05) |
| | 5 | 6.65E-02 (4.82E-03) | 5.05E-02 (3.16E-03) | 2.67E-02 (5.49E-03) | 1.27E-01 (7.78E-05) | 1.09E-01 (4.31E-03) | 1.17E-01 (4.01E-05) |
| | 8 | 4.37E-02 (2.17E-02) | 1.22E-04 (1.59E-04) | 1.55E-03 (2.14E-03) | 1.04E-01 (1.13E-04) | 3.64E-02 (2.62E-02) | 9.59E-02 (3.72E-06) |
| | 10 | 8.55E-02 (1.08E-02) | 1.22E-05 (1.73E-05) | 1.04E-04 (1.27E-04) | 9.98E-02 (2.66E-04) | 6.92E-06 (9.79E-06) | 9.39E-02 (6.03E-05) |
| | 15 | 4.53E-02 (3.58E-02) | 7.77E-10 (6.77E-10) | 4.11E-08 (5.81E-08) | 9.43E-02 (4.16E-04) | 4.23E-02 (2.43E-02) | 9.21E-02 (8.31E-05) |
| DTLZ6 | 3 | 1.86E-01 (4.95E-05) | 1.86E-01 (2.33E-04) | 1.86E-01 (2.51E-03) | 1.87E-01 (0.00E+0) | 1.77E-01 (1.15E-03) | 2.00E-01 (6.27E-05) |
| | 5 | 0.00E+0 (0.00E+0) | 0.00E+0 (0.00E+0) | 0.00E+0 (0.00E+0) | 1.27E-01 (3.25E-04) | 0.00E+0 (0.00E+0) | 0.00E+00 (8.00E-06) |
| | 8 | 0.00E+0 (0.00E+0) | 0.00E+0 (0.00E+0) | 0.00E+0 (0.00E+0) | 1.04E-01 (9.19E-05) | 0.00E+0 (0.00E+0) | 0.00E+00 (1.00E-05) |
| | 10 | 0.00E+0 (0.00E+0) | 0.00E+0 (0.00E+0) | 0.00E+0 (0.00E+0) | 9.98E-02 (2.02E-04) | 0.00E+0 (0.00E+0) | 0.00E+00 (3.54E-05) |
| | 15 | 0.00E+0 (0.00E+0) | 0.00E+0 (0.00E+0) | 0.00E+0 (0.00E+0) | 9.43E-02 (7.78E-06) | 0.00E+0 (0.00E+0) | 0.00E+00 (0.00E+00) |
| DTLZ7 | 3 | 2.76E-01 (2.12E-05) | 2.76E-01 (7.78E-05) | 2.77E-01 (6.51E-04) | 2.63E-01 (8.98E-04) | 2.77E-01 (1.46E-03) | 2.76E-01 (5.67E-05) |
| | 5 | 2.47E-01 (1.56E-03) | 2.48E-01 (7.57E-04) | 2.45E-01 (1.34E-04) | 1.41E-01 (5.09E-04) | 2.20E-01 (9.77E-03) | 2.14E-01 (5.08E-05) |
| | 8 | 1.83E-01 (1.56E-04) | 1.41E-01 (7.97E-03) | 1.42E-01 (5.87E-03) | 4.18E-04 (4.94E-04) | 3.32E-02 (3.46E-02) | 3.45E-02 (4.22E-05) |
| | 10 | 1.75E-01 (2.88E-03) | -4.24E-01 (8.14E-01) | 1.29E-01 (8.29E-03) | 1.29E-04 (1.28E-04) | 3.14E-05 (4.33E-05) | 4.04E-02 (5.90E-05) |
| | 15 | 1.46E-01 (1.62E-03) | 8.99E-02 (2.15E-03) | 1.21E-01 (1.77E-02) | 3.24E-06 (4.29E-06) | 2.07E-11 (1.75E-12) | 2.47E-08 (8.25E-05) |
| +/ = /- | | | 18/4/13 | 18/5/12 | 13/0/22 | 16/6/13 | 21/7/7 |

In Fig 7, the approximated PF obtained by the competing algorithms for the series-parallel system is presented to further explain the results.

## 5.2 Overspeed protection for gas turbine problem

This system comprises of a fuel-supplied gas turbine through various valves. Fig 8 depicts a four-valved overspeed prevention system for gas turbines. The valves regulate the fuel flow when overspeed is detected. The problem can be expressed mathematically in the following

**Table 10. The performance metrics comparison between MaAVOA and other algorithms in terms of IGD value on DTLZs in case of the computational time is 30 seconds.**

| | | MAAVOA | NSGA3 | UNSGA3 | MOEAD | CTAEA | AGEMOEA |
|---|---|---|---|---|---|---|---|
| PROB. | OBJ. | IGD | IGD | IGD | IGD | IGD | IGD |
| DTLZ1 | 3 | 1.54E-2 (6.36E-6) | 1.54E-2 (7.07E-7) | 1.54E-2 (7.07E-7) | 1.83E-2 (2.69E-4) | 1.60E-2 (4.02E-4) | 3.95E-02 (6.51E-5) |
| | 5 | 5.35E-2 (3.49E-4) | 5.27E-2 (1.34E-5) | 5.30E-2 (3.80E-4) | 5.43E-2 (3.07E-3) | 1.34E+0 (2.68E-1) | 5.16E-01 (7.15E-5) |
| | 8 | 1.22E-1 (3.82E-4) | 1.22E-1 (2.55E-4) | 1.22E-1 (5.16E-4) | 1.91E-1 (1.37E-1) | 4.78E-1 (1.05E-1) | 2.38E-01 (4.09E-04) |
| | 10 | 2.04E-1 (3.40E-3) | 1.44E-1 (3.08E-3) | 1.43E-1 (1.13E-3) | 1.13E-1 (1.86E-3) | 1.70E+0 (1.10E+0) | 9.56E-01 (4.74E-5) |
| | 15 | 4.84E-1 (2.64E-1) | 2.21E-1 (4.03E-4) | 2.23E-1 (7.14E-4) | 1.57E-1 (7.67E-3) | 3.98E-1 (3.20E-1) | 1.46E+00 (1.57E-5) |
| DTLZ2 | 3 | 4.09E-2 (6.36E-6) | 4.09E-2 (4.24E-6) | 4.09E-2 (4.24E-6) | 4.10E-2 (5.37E-5) | 4.17E-2 (9.97E-5) | 5.34E-02 (3.97E-5) |
| | 5 | 1.65E-1 (5.66E-5) | 1.65E-1 (9.90E-5) | 1.65E-1 (7.07E-6) | 1.65E-1 (1.27E-4) | 1.97E-1 (4.04E-3) | 2.44E-01 (5.17E-5) |
| | 8 | 3.60E-1 (8.49E-5) | 3.59E-1 (1.98E-4) | 3.59E-1 (1.77E-4) | 3.57E-1 (6.58E-4) | 3.90E-1 (1.30E-2) | 4.70E-01 (4.47E-5) |
| | 10 | 4.61E-1 (3.54E-3) | 4.61E-1 (4.95E-5) | 4.62E-1 (1.39E-3) | 4.32E-1 (8.49E-3) | 5.12E-1 (4.13E-3) | 5.95E-01 (9.68E-5) |
| | 15 | 6.36E-1 (2.47E-4) | 6.34E-1 (4.74E-4) | 6.34E-1 (9.69E-4) | 8.47E-1 (2.24E-1) | 5.69E-1 (3.91E-3) | 6.93E-01 (9.36E-5) |
| DTLZ3 | 3 | 4.10E-2 (6.08E-5) | 4.10E-2 (6.36E-5) | 4.10E-2 (5.66E-5) | 3.80E-1 (2.14E-2) | 7.21E-1 (6.48E-1) | 4.99E-02 (5.74E-5) |
| | 5 | 1.65E-1 (1.39E-3) | 1.76E-1 (1.50E-2) | 1.66E-1 (3.89E-4) | 1.14E+1 (1.45E+1) | 6.12E+1 (5.43E+1) | 2.06E+01 (6.06E-5) |
| | 8 | 4.41E-1 (3.61E-2) | 3.65E-1 (3.25E-3) | 3.64E-1 (7.78E-5) | 2.57E+0 (4.54E-1) | 3.09E+1 (2.39E+1) | 9.95E+00 (3.87E-5) |
| | 10 | 2.78E+0 (2.60E+0) | 4.86E-1 (2.58E-2) | 4.79E-1 (9.98E-3) | 1.97E+0 (1.00E-1) | 9.67E+1 (2.79E+1) | 4.15E+01 (2.48E-5) |
| | 15 | 7.94E+0 (8.74E+0) | 6.58E-1 (1.00E-2) | 6.74E-1 (5.24E-2) | 2.19E+0 (1.23E+0) | 1.48E+1 (8.04E-1) | 1.65E+01 (5.02E-5) |
| DTLZ4 | 3 | 4.09E-2 (4.03E-5) | 4.09E-2 (1.41E-6) | 4.09E-2 (0.00E+0) | 2.91E-1 (3.54E-1) | 4.19E-2 (2.09E-4) | 5.34E-02 (6.82E-5) |
| | 5 | 1.65E-1 (1.48E-4) | 1.65E-1 (3.54E-5) | 1.65E-1 (0.00E+0) | 8.67E-1 (6.36E-5) | 1.88E-1 (1.39E-3) | 2.50E-01 (5.87E-5) |
| | 8 | 3.59E-1 (3.11E-4) | 3.59E-1 (3.54E-5) | 3.59E-1 (3.54E-5) | 6.24E-1 (2.27E-1) | 3.81E-1 (1.66E-3) | 4.39E-01 (4.49E-5) |
| | 10 | 4.63E-1 (3.44E-3) | 4.58E-1 (5.94E-4) | 4.58E-1 (6.65E-4) | 8.43E-1 (6.51E-2) | 6.27E-1 (1.66E-2) | 5.35E-01 (5.98E-5) |
| | 15 | 6.33E-1 (7.07E-6) | 6.32E-1 (4.95E-5) | 6.32E-1 (7.07E-6) | 9.29E-1 (1.50E-1) | 7.00E-1 (2.55E-3) | 5.86E-01 (6.09E-5) |
| DTLZ5 | 3 | 2.71E-2 (2.09E-3) | 2.80E-2 (1.56E-3) | 2.88E-2 (1.46E-4) | 2.49E-2 (3.74E-4) | 5.21E-3 (5.18E-4) | 6.90E-03 (8.23E-5) |
| | 5 | 1.19E-1 (6.60E-3) | 1.35E-1 (8.60E-3) | 1.37E-1 (3.85E-2) | 2.24E-2 (7.86E-4) | 1.08E-1 (1.05E-2) | 1.22E-01 (2.55E-5) |
| | 8 | 1.38E-1 (2.51E-2) | 3.01E-1 (1.42E-3) | 3.61E-1 (9.77E-2) | 6.89E-2 (1.80E-4) | 4.63E-1 (2.01E-2) | 2.19E-01 (3.21E-5) |
| | 10 | 1.76E-1 (6.23E-2) | 4.09E-1 (2.42E-2) | 3.49E-1 (6.26E-3) | 6.64E-2 (3.30E-4) | 5.25E-1 (5.97E-2) | 3.96E-01 (6.60E-5) |
| | 15 | 1.50E-1 (3.66E-2) | 3.92E-1 (1.44E-1) | 3.89E-1 (5.53E-2) | 1.70E-1 (8.61E-3) | 3.09E-1 (6.68E-2) | 3.30E-01 (5.32E-5) |
| DTLZ6 | 3 | 2.78E-2 (2.48E-3) | 2.72E-2 (1.22E-3) | 2.93E-2 (1.64E-3) | 5.18E-1 (4.76E-1) | 1.23E+0 (1.25E-1) | 5.99E-03 (8.94E-5) |
| | 5 | 2.28E+0 (7.66E-1) | 1.24E+0 (4.64E-1) | 1.21E+0 (4.45E-1) | 3.76E+0 (5.99E-1) | 7.80E+0 (3.52E-1) | 7.84E+00 (1.67E-5) |
| | 8 | 3.44E+0 (6.68E-1) | 2.00E+0 (4.21E-1) | 2.06E+0 (6.39E-1) | 3.48E+0 (5.86E-1) | 7.45E+0 (1.55E-1) | 9.56E+00 (5.66E-5) |
| | 10 | 6.21E+0 (6.67E-2) | 5.38E+0 (2.92E-1) | 5.67E+0 (4.54E-1) | 1.24E+0 (1.65E+0) | 8.19E+0 (2.14E-1) | 9.92E+00 (7.49E-5) |
| | 15 | 2.31E+0 (3.26E-1) | 1.54E+0 (1.71E-1) | 1.42E+0 (6.44E-2) | 3.55E-1 (1.89E-1) | 7.31E+0 (2.13E-1) | 9.87E+00 (9.85E-5) |
| DTLZ7 | 3 | 7.29E-2 (1.40E-4) | 7.27E-2 (7.16E-4) | 7.39E-2 (6.53E-4) | 1.05E-1 (2.97E-3) | 6.10E-2 (4.84E-4) | 4.26E-02 (9.19E-5) |
| | 5 | 3.75E-1 (1.03E-3) | 3.64E-1 (8.51E-3) | 3.53E-1 (6.35E-3) | 6.38E-1 (1.44E-1) | 2.79E-1 (7.78E-5) | 2.61E-01 (7.21E-5) |
| | 8 | 1.18E+0 (2.69E-3) | 1.22E+0 (6.86E-2) | 1.30E+0 (2.69E-2) | 4.02E+0 (2.64E+0) | 1.97E+0 (4.22E-1) | 8.01E-01 (6.46E-5) |
| | 10 | 2.16E+0 (1.75E-1) | 1.96E+0 (3.35E-2) | 1.96E+0 (6.73E-2) | 3.45E+0 (1.35E+0) | 1.75E+1 (1.74E+0) | 1.17E+00 (6.15E-5) |
| | 15 | 8.70E+0 (5.44E-2) | 8.36E+0 (1.96E-2) | 8.38E+0 (1.04E-1) | 3.22E+0 (6.00E-2) | 1.88E+1 (8.37E+0) | 2.27E+00 (1.80E-5) |
| +/ = /- | | | **6/5/24** | **6/5/24** | **22/5/8** | **27/5/3** | **8/0/27** |

way:

$$max\, f(r, n) = \prod_{i=1}^{m}[1 - (1 - r_i)^{n_i}] \tag{30}$$

$$min\, V_s(n) = \sum_{i=m}^{m} w_i v_i^2 n_i^2 \tag{31}$$

$$min\, C_s(r, n) = \sum_{i=1}^{m} C(r_i)[n_i + exp(0.25 n_i)] \tag{32}$$

**Table 11. The performance metrics comparison between MaAVOA and other algorithms in terms of GD value on DTLZs in case of the computational time is 30 seconds.**

| | | MAAVOA | NSGA3 | UNSGA3 | MOEAD | CTAEA | AGEMOEA |
|---|---|---|---|---|---|---|---|
| PROB. | OBJ. | GD | GD | GD | GD | GD | GD |
| DTLZ1 | 3 | 1.48E-4 (7.92E-7) | 1.48E-4 (1.98E-7) | 1.48E-4 (2.40E-7) | 7.35E-4 (6.27E-5) | 2.66E-4 (1.21E-4) | 4.06E-02 (5.81E-05) |
| | 5 | 1.02E-3 (6.36E-6) | 1.07E-3 (4.24E-7) | 1.08E-3 (9.40E-6) | 1.35E-3 (1.07E-4) | 8.82E-1 (1.66E-1) | 9.55E+01 (6.43E-05) |
| | 8 | 3.28E-3 (3.58E-4) | 2.98E-3 (1.91E-6) | 2.99E-3 (2.05E-6) | 1.26E-2 (1.11E-2) | 9.90E-2 (4.53E-3) | 2.89E+00 (2.82E-05) |
| | 10 | 9.38E-3 (1.04E-3) | 1.10E-3 (9.02E-4) | 6.41E-4 (7.58E-5) | 2.73E-3 (3.96E-5) | 1.51E+0 (1.34E-2) | 1.46E+02 (3.91E-05) |
| | 15 | 9.62E-2 (1.15E-1) | 9.42E-4 (8.11E-4) | 3.95E-4 (3.11E-5) | 7.08E-3 (5.24E-4) | 7.54E-1 (6.25E-1) | 1.10E+02 (8.72E-05) |
| DTLZ2 | 3 | 3.96E-4 (2.13E-6) | 3.97E-4 (2.38E-6) | 3.97E-4 (2.67E-6) | 4.33E-4 (1.59E-5) | 4.05E-4 (7.69E-6) | 4.10E-02 (1.83E-05) |
| | 5 | 3.39E-3 (8.34E-6) | 3.41E-3 (1.11E-5) | 3.41E-3 (6.86E-6) | 3.43E-3 (5.18E-5) | 6.89E-3 (4.89E-4) | 2.43E-01 (2.57E-05) |
| | 8 | 9.64E-3 (2.69E-5) | 9.63E-3 (2.83E-7) | 9.64E-3 (6.36E-6) | 8.94E-3 (1.16E-4) | 1.04E-2 (1.83E-3) | 8.71E-01 (4.97E-05) |
| | 10 | 3.53E-3 (9.14E-5) | 1.81E-3 (1.13E-4) | 2.12E-3 (5.16E-4) | 8.49E-3 (3.96E-4) | 2.40E-2 (8.03E-4) | 1.04E+00 (8.10E-05) |
| | 15 | 2.31E-3 (1.47E-4) | 8.80E-4 (4.30E-4) | 1.40E-3 (2.76E-4) | 1.65E-2 (6.72E-3) | 1.90E-2 (1.27E-3) | 1.55E+00 (4.91E-05) |
| DTLZ3 | 3 | 4.03E-4 (1.26E-5) | 4.07E-4 (1.51E-5) | 4.08E-4 (1.16E-5) | 6.61E-2 (1.76E-2) | 1.63E-1 (4.21E-2) | 3.87E-02 (6.17E-05) |
| | 5 | 3.46E-3 (6.47E-5) | 6.59E-3 (4.47E-3) | 3.41E-3 (1.28E-5) | 1.13E+0 (1.48E+0) | 1.52E+1 (1.01E+1) | 1.25E+02 (4.94E-05) |
| | 8 | 4.51E-2 (4.06E-2) | 9.87E-3 (5.54E-5) | 1.38E-2 (1.24E-3) | 1.70E-1 (1.02E-2) | 8.76E+0 (4.01E+0) | 1.87E+02 (3.56E-05) |
| | 10 | 4.08E-1 (3.41E-1) | 1.51E-2 (8.44E-3) | 1.15E-2 (3.11E-3) | 8.30E-2 (5.75E-3) | 1.90E+1 (5.30E-1) | 4.14E+02 (6.14E-05) |
| | 15 | 1.63E+0 (1.93E+0) | 9.67E-3 (2.86E-3) | 4.14E-2 (4.23E-2) | 1.11E-1 (1.52E-1) | 5.81E+0 (1.36E+0) | 2.95E+02 (8.39E-05) |
| DTLZ4 | 3 | 4.64E-4 (9.52E-5) | 3.96E-4 (1.91E-7) | 3.97E-4 (4.24E-7) | 3.48E-4 (1.31E-4) | 3.94E-4 (2.21E-6) | 3.99E-02 (9.96E-05) |
| | 5 | 3.43E-3 (5.78E-5) | 3.41E-3 (4.10E-6) | 3.40E-3 (7.35E-6) | 1.80E-3 (1.00E-4) | 5.12E-3 (1.73E-4) | 3.41E-01 (5.98E-05) |
| | 8 | 1.17E-2 (2.88E-3) | 9.64E-3 (1.36E-5) | 9.64E-3 (1.20E-5) | 5.50E-3 (2.33E-3) | 9.53E-3 (2.24E-4) | 1.10E+00 (8.82E-05) |
| | 10 | 6.07E-3 (4.36E-3) | 1.33E-3 (1.57E-4) | 1.21E-3 (8.97E-5) | 2.22E-3 (1.16E-3) | 3.25E-2 (3.48E-4) | 1.30E+00 (3.12E-05) |
| | 15 | 1.23E-2 (2.31E-3) | 7.92E-5 (2.41E-5) | 1.61E-4 (3.22E-5) | 6.37E-3 (5.75E-3) | 2.60E-2 (2.07E-3) | 1.86E+00 (9.90E-05) |
| DTLZ5 | 3 | 5.00E-5 (5.27E-5) | 1.16E-3 (7.55E-5) | 6.31E-4 (1.63E-4) | 3.26E-3 (4.61E-3) | 3.16E-4 (2.33E-4) | 2.85E-03 (4.44E-05) |
| | 5 | 2.02E-1 (1.96E-2) | 2.11E-1 (3.50E-3) | 1.99E-1 (8.66E-3) | 8.57E-3 (1.52E-3) | 9.78E-2 (1.58E-3) | 2.01E+00 (2.06E-05) |
| | 8 | 2.12E-1 (1.18E-3) | 1.63E-1 (2.58E-2) | 1.45E-1 (1.10E-2) | 6.10E-4 (3.53E-4) | 1.15E-1 (3.63E-3) | 4.80E+00 (9.06E-05) |
| | 10 | 1.81E-1 (1.27E-2) | 1.58E-1 (6.72E-4) | 2.09E-1 (4.20E-2) | 1.47E-6 (1.11E-7) | 9.52E-2 (2.65E-3) | 6.34E+00 (9.93E-05) |
| | 15 | 1.09E-1 (1.25E-1) | 1.47E-1 (1.21E-2) | 1.70E-1 (3.18E-2) | 1.47E-6 (4.72E-7) | 1.26E-1 (1.98E-4) | 1.53E+01 (8.78E-05) |
| DTLZ6 | 3 | 9.68E-6 (2.44E-6) | 1.09E-5 (3.36E-7) | 1.09E-5 (5.95E-7) | 7.24E-2 (4.98E-2) | 1.61E-1 (1.58E-2) | 4.93E-05 (6.30E-05) |
| | 5 | 2.57E-1 (6.17E-2) | 2.63E-1 (3.24E-2) | 2.42E-1 (5.86E-2) | 3.19E-1 (2.34E-2) | 6.07E-1 (1.45E-2) | 2.04E+01 (9.53E-05) |
| | 8 | 3.49E-1 (4.23E-2) | 2.78E-1 (3.00E-2) | 3.07E-1 (4.17E-2) | 3.27E-1 (4.46E-2) | 6.82E-1 (5.80E-4) | 4.78E+01 (3.09E-05) |
| | 10 | 4.55E-1 (3.55E-3) | 3.98E-1 (3.71E-2) | 4.07E-1 (4.88E-2) | 1.01E-1 (1.42E-1) | 5.53E-1 (2.28E-3) | 7.92E+01 (7.31E-05) |
| | 15 | 3.64E-1 (5.10E-2) | 2.34E-1 (2.07E-2) | 2.28E-1 (1.50E-2) | 1.14E-6 (2.99E-7) | 7.00E-1 (9.43E-3) | 2.71E+02 (1.10E-05) |
| DTLZ7 | 3 | 9.49E-3 (1.01E-2) | 2.14E-3 (4.25E-4) | 2.27E-3 (6.24E-5) | 2.94E-3 (1.84E-4) | 2.48E-3 (6.09E-5) | 1.67E-02 (8.64E-05) |
| | 5 | 9.89E-3 (1.01E-3) | 1.39E-2 (1.05E-4) | 1.29E-2 (6.10E-4) | 4.33E-3 (6.55E-5) | 1.69E-2 (3.99E-3) | 1.74E-01 (5.04E-05) |
| | 8 | 8.12E-2 (2.98E-2) | 1.86E-1 (2.28E-1) | 8.22E-2 (7.58E-2) | 2.11E-2 (8.94E-3) | 2.35E-1 (4.82E-3) | 6.01E-01 (8.83E-05) |
| | 10 | 2.00E-2 (1.78E-3) | 1.93E-1 (9.13E-2) | 1.45E-1 (1.39E-3) | 1.59E-2 (4.45E-4) | 2.12E+0 (5.46E-2) | 1.08E+00 (1.45E-05) |
| | 15 | 7.76E-2 (1.81E-3) | 2.40E-1 (5.19E-2) | 2.43E-1 (3.51E-2) | 9.81E-2 (7.21E-3) | 4.16E+0 (1.49E-1) | 2.49E+00 (7.52E-05) |
| +/ = /- | | | 8/5/21 | 9/5/21 | 20/5/10 | 30/5/0 | 0/0/35 |

$$min\ W_s(n) = w_i n_i exp\left(0.25 n_i\right) \qquad (33)$$

$$s.t.\ V_s - V \leq 0, C_s - C \leq 0, W_s - W \leq 0\ with\ 0.5 \leq r_i \geq 10^{-6}, \quad r_i \in \mathbb{R}^+, n_i \in \mathbb{Z}^+, 1 \leq n_i \geq 10$$

$$where\ C(r_i) = \alpha_i \left(-\frac{T}{\log(r_i)}\right)^{\beta_i}$$

**Table 12. The performance metrics comparison between MaAVOA and other algorithms in terms of HV value on DTLZs in case of the computational time is 30 seconds.**

| PROB. | OBJ. | MAAVOA | NSGA3 | UNSGA3 | MOEAD | CTAEA | AGEMOEA |
|---|---|---|---|---|---|---|---|
| | | HV | HV | HV | HV | HV | HV |
| DTLZ1 | 3 | 8.50E-1 (1.13E-4) | 8.50E-1 (2.83E-5) | 8.50E-1 (1.41E-5) | 8.36E-1 (1.23E-3) | 8.48E-1 (1.08E-3) | 8.40E-01 (7.04E-5) |
| | 5 | 9.78E-1 (8.49E-5) | 9.80E-1 (2.05E-4) | 9.79E-1 (4.31E-4) | 9.57E-1 (5.13E-3) | 0.00E+0 (0.00E+0) | 7.01E-01 (6.56E-5) |
| | 8 | 9.97E-1 (1.34E-4) | 9.97E-1 (9.90E-5) | 9.97E-1 (4.95E-5) | 8.15E-1 (1.95E-1) | 1.24E-1 (1.30E-1) | 8.92E-01 (5.75E-5) |
| | 10 | 9.63E-1 (2.75E-3) | 9.99E-1 (5.16E-4) | 9.99E-1 (6.36E-5) | 8.12E-1 (1.35E-2) | 0.00E+0 (0.00E+0) | 1.11E-01 (1.90E-5) |
| | 15 | 3.99E-1 (5.61E-1) | 1.00E+0 (3.54E-5) | 1.00E+0 (2.83E-5) | 5.54E-1 (3.01E-2) | 4.21E-1 (5.76E-1) | 8.28E-03 (3.90E-5) |
| DTLZ2 | 3 | 5.71E-1 (1.06E-4) | 5.71E-1 (8.49E-5) | 5.71E-1 (2.12E-5) | 5.68E-1 (6.01E-4) | 5.69E-1 (2.83E-4) | 5.55E-01 (5.89E-5) |
| | 5 | 8.10E-1 (8.84E-4) | 8.12E-1 (5.09E-4) | 8.12E-1 (7.07E-6) | 7.98E-1 (4.88E-4) | 7.34E-1 (9.47E-3) | 6.84E-01 (5.11E-5) |
| | 8 | 9.32E-1 (6.08E-4) | 9.35E-1 (3.46E-4) | 9.35E-1 (4.60E-4) | 9.29E-1 (1.44E-3) | 8.56E-1 (2.49E-2) | 6.07E-01 (2.51E-5) |
| | 10 | 9.62E-1 (2.28E-3) | 9.69E-1 (9.05E-4) | 9.68E-1 (2.17E-3) | 9.43E-1 (8.56E-3) | 6.46E-1 (8.62E-2) | 5.81E-01 (9.11E-5) |
| | 15 | 9.87E-1 (3.39E-4) | 9.89E-1 (6.65E-4) | 9.89E-1 (6.22E-4) | 5.39E-1 (2.39E-1) | 8.05E-1 (4.02E-3) | 2.64E-01 (8.08E-5) |
| DTLZ3 | 3 | 5.70E-1 (1.22E-3) | 5.70E-1 (1.23E-3) | 5.69E-1 (9.62E-4) | 1.03E-1 (2.75E-2) | 9.33E-2 (1.32E-1) | 5.56E-01 (6.32E-5) |
| | 5 | 8.02E-1 (9.81E-3) | 7.97E-1 (1.38E-2) | 8.05E-1 (5.47E-3) | 0.00E+0 (0.00E+0) | 0.00E+0 (0.00E+0) | 0.00E+0 (0.00E+0) |
| | 8 | 7.83E-1 (3.75E-2) | 9.18E-1 (9.88E-3) | 9.24E-1 (1.41E-5) | 0.00E+0 (0.00E+0) | 0.00E+0 (0.00E+0) | 0.00E+0 (0.00E+0) |
| | 10 | 4.22E-2 (5.97E-2) | 8.93E-1 (5.75E-2) | 9.16E-1 (1.31E-2) | 0.00E+0 (0.00E+0) | 0.00E+0 (0.00E+0) | 0.00E+0 (0.00E+0) |
| | 15 | 0.00E+0 (0.00E+0) | 9.63E-1 (1.72E-2) | 8.88E-1 (1.27E-1) | 2.45E-2 (3.47E-2) | 0.00E+0 (0.00E+0) | 0.00E+0 (0.00E+0) |
| DTLZ4 | 3 | 5.71E-1 (1.27E-4) | 5.71E-1 (3.54E-5) | 5.71E-1 (0.00E+0) | 4.57E-1 (1.58E-1) | 5.69E-1 (8.49E-5) | 5.52E-01 (7.95E-5) |
| | 5 | 8.11E-1 (1.27E-3) | 8.12E-1 (1.20E-4) | 8.12E-1 (9.90E-5) | 3.24E-1 (1.94E-3) | 7.60E-1 (5.53E-3) | 7.00E-01 (2.38E-5) |
| | 8 | 9.35E-1 (9.90E-5) | 9.35E-1 (7.78E-5) | 9.36E-1 (1.91E-4) | 7.71E-1 (1.81E-1) | 8.92E-1 (1.77E-3) | 7.29E-01 (5.75E-5) |
| | 10 | 9.65E-1 (3.75E-3) | 9.72E-1 (2.47E-4) | 9.73E-1 (3.18E-4) | 6.97E-1 (7.99E-2) | 5.95E-1 (5.02E-2) | 7.55E-01 (9.14E-5) |
| | 15 | 9.90E-1 (4.95E-5) | 9.90E-1 (2.12E-5) | 9.90E-1 (1.06E-4) | 7.28E-1 (2.10E-1) | 8.33E-1 (1.18E-3) | 7.64E-01 (2.75E-5) |
| DTLZ5 | 3 | 1.86E-1 (1.56E-4) | 1.85E-1 (1.63E-4) | 1.85E-1 (2.83E-4) | 1.87E-1 (7.78E-5) | 1.99E-1 (1.48E-4) | 1.98E-01 (6.29E-5) |
| | 5 | 7.27E-2 (1.69E-2) | 3.51E-2 (6.48E-3) | 5.79E-2 (4.07E-2) | 1.15E-1 (6.94E-3) | 7.72E-2 (4.65E-3) | 1.17E-01 (2.34E-5) |
| | 8 | 2.24E-2 (1.08E-2) | 6.06E-5 (2.31E-5) | 5.54E-4 (7.83E-4) | 1.04E-1 (1.41E-4) | 6.70E-5 (9.47E-5) | 9.71E-02 (3.02E-5) |
| | 10 | 4.21E-2 (5.75E-2) | 0.00E+0 (0.00E+0) | 4.27E-5 (3.67E-5) | 1.00E-1 (2.68E-4) | 1.13E-8 (1.59E-8) | 9.40E-02 (7.53E-5) |
| | 15 | 9.08E-2 (1.18E-3) | 0.00E+0 (0.00E+0) | 1.15E-7 (1.59E-7) | 9.39E-2 (1.82E-4) | 1.95E-6 (2.76E-6) | 9.17E-02 (4.60E-5) |
| DTLZ6 | 3 | 1.86E-1 (6.58E-4) | 1.86E-1 (2.19E-4) | 1.86E-1 (3.11E-4) | 2.13E-2 (3.01E-2) | 0.00E+0 (0.00E+0) | 2.00E-01 (3.99E-5) |
| | 5 | 0.00E+0 (0.00E+0) | 0.00E+0 (0.00E+0) | 0.00E+0 (0.00E+0) | 0.00E+0 (0.00E+0) | 0.00E+0 (0.00E+0) | 0.00E+00 (1.17E-5) |
| | 8 | 0.00E+0 (0.00E+0) | 0.00E+0 (0.00E+0) | 0.00E+0 (0.00E+0) | 0.00E+0 (0.00E+0) | 0.00E+0 (0.00E+0) | 0.00E+00 (3.81E-5) |
| | 10 | 0.00E+0 (0.00E+0) | 0.00E+0 (0.00E+0) | 0.00E+0 (0.00E+0) | 5.00E-2 (7.07E-2) | 0.00E+0 (0.00E+0) | 0.00E+00 (8.85E-5) |
| | 15 | 0.00E+0 (0.00E+0) | 0.00E+0 (0.00E+0) | 0.00E+0 (0.00E+0) | 9.43E-2 (2.08E-4) | 0.00E+0 (0.00E+0) | 0.00E+00 (9.45E-5) |
| DTLZ7 | 3 | 2.76E-1 (4.31E-4) | 2.76E-1 (1.70E-4) | 2.76E-1 (1.48E-4) | 2.60E-1 (7.99E-4) | 2.76E-1 (3.11E-4) | 2.76E-01 (2.08E-5) |
| | 5 | 2.48E-1 (1.17E-3) | 2.46E-1 (1.70E-4) | 2.46E-1 (7.71E-4) | 6.53E-2 (1.70E-2) | 1.89E-1 (4.50E-3) | 1.96E-01 (8.55E-5) |
| | 8 | 1.82E-1 (1.91E-3) | 1.61E-1 (7.44E-3) | 1.47E-1 (2.35E-2) | 3.59E-4 (5.03E-4) | 2.54E-5 (1.07E-5) | 4.87E-02 (7.88E-5) |
| | 10 | 1.73E-1 (6.51E-3) | 1.47E-1 (2.67E-3) | 1.27E-1 (2.10E-2) | 1.17E-4 (1.52E-4) | 0.00E+0 (0.00E+0) | 3.31E-03 (9.27E-5) |
| | 15 | 1.33E-1 (7.28E-3) | 7.15E-2 (2.09E-2) | 1.08E-1 (3.59E-2) | 3.91E-6 (3.57E-6) | 1.86E-13 (2.62E-13) | 2.75E-11 (6.31E-5) |
| +/ = /- | | | **11/5/19** | **11/5/19** | **24/4/8** | **27/6/3** | **23/6/6** |

$\alpha_i$ and $\beta_i$ are constants representing the actual features of each item at stage $i$ and $T$ is the operating time during which the item should not fail. Table 16 provides the input data for an overspeed protection for gas turbine system.

The algorithms are terminated after 250, 500, 1000, 2000, 4000, 5000, and 10000 generations. The engineering problem has 4 objective functions. Accordingly, the population size is chosen to be 969 ($Nr_1 = 16$, $Nr_2 = 0$, and $nRef = 969$).

The results for the overspeed protection for gas turbine problem are given in the Table 17 and Fig 9.

**Table 13. Comparison between MaAVOA and other algorithms in terms of the number of generation and number of function evaluations on DTLZs in case of the computational time is 30 seconds.**

| PROB. | OBJ. | MAAVOA | | NSGA3 | | UNSGA3 | | MOEAD | | CTAEA | | AGEMOEA | |
|---|---|---|---|---|---|---|---|---|---|---|---|---|---|
| | | N_GEN | N_EVAL | N_GEN | N_EVAL | N_GEN | N_EVAL | N_GEN | N_EVAL | N_GEN | N_EVAL | N_GEN | N_EVAL |
| DTLZ1 | 3 | 460 | 87955 | 1050 | 160650 | 1037 | 158661 | 148 | 22644 | 210 | 32130 | 718 | 71800 |
| | 5 | 275 | 75345 | 678 | 142380 | 667 | 140070 | 106 | 22260 | 91 | 19110 | 491 | 49100 |
| | 8 | 393 | 82653 | 815 | 127140 | 810 | 126360 | 139 | 21684 | 141 | 21996 | 618 | 61800 |
| | 10 | 223 | 83613 | 304 | 83600 | 300 | 82500 | 58 | 15950 | 43 | 11825 | 401 | 40100 |
| | 15 | 609 | 113498 | 724 | 97740 | 722 | 97470 | 128 | 17280 | 124 | 16740 | 325 | 32500 |
| DTLZ2 | 3 | 321 | 66209 | 936 | 143208 | 837 | 128061 | 141 | 21573 | 137 | 20961 | 556 | 55600 |
| | 5 | 214 | 61471 | 598 | 125580 | 585 | 122850 | 97 | 20370 | 66 | 13860 | 412 | 41200 |
| | 8 | 272 | 58549 | 706 | 110136 | 748 | 116688 | 129 | 20124 | 78 | 12168 | 325 | 32500 |
| | 10 | 100 | 38009 | 277 | 76175 | 279 | 76725 | 63 | 17325 | 27 | 7425 | 302 | 30200 |
| | 15 | 260 | 48692 | 676 | 91260 | 672 | 90720 | 129 | 17415 | 83 | 11205 | 264 | 26400 |
| DTLZ3 | 3 | 530 | 109354 | 1091 | 166923 | 1055 | 161415 | 157 | 24021 | 277 | 42381 | 817 | 81700 |
| | 5 | 305 | 87598 | 687 | 144270 | 673 | 141330 | 95 | 19950 | 72 | 15120 | 560 | 56000 |
| | 8 | 404 | 87050 | 869 | 135564 | 844 | 131664 | 130 | 20280 | 138 | 21528 | 476 | 47600 |
| | 10 | 228 | 86780 | 301 | 82775 | 333 | 91575 | 63 | 17325 | 42 | 11550 | 312 | 31200 |
| | 15 | 601 | 112568 | 743 | 100305 | 731 | 98685 | 129 | 17415 | 116 | 15660 | 308 | 30800 |
| DTLZ4 | 3 | 428 | 88364 | 1097 | 167841 | 1064 | 162792 | 158 | 24174 | 139 | 21267 | 571 | 57100 |
| | 5 | 212 | 60858 | 594 | 124740 | 576 | 120960 | 101 | 21210 | 64 | 13440 | 422 | 42200 |
| | 8 | 262 | 56422 | 740 | 115440 | 709 | 110604 | 130 | 20280 | 78 | 12168 | 327 | 32700 |
| | 10 | 82 | 31173 | 250 | 68750 | 256 | 70400 | 50 | 13750 | 25 | 6875 | 255 | 25500 |
| | 15 | 220 | 41218 | 657 | 88695 | 650 | 87750 | 125 | 16875 | 79 | 10665 | 233 | 23300 |
| DTLZ5 | 3 | 912 | 188430 | 1270 | 194310 | 1206 | 184518 | 150 | 22950 | 358 | 54774 | 631 | 63100 |
| | 5 | 495 | 142285 | 657 | 137970 | 642 | 134820 | 96 | 20160 | 154 | 32340 | 450 | 45000 |
| | 8 | 669 | 144065 | 834 | 130104 | 788 | 122928 | 127 | 19812 | 159 | 24804 | 257 | 25700 |
| | 10 | 291 | 110875 | 292 | 80300 | 318 | 87450 | 62 | 17050 | 69 | 18975 | 314 | 31400 |
| | 15 | 658 | 123247 | 728 | 98280 | 717 | 96795 | 129 | 17415 | 165 | 22275 | 284 | 28400 |
| DTLZ6 | 3 | 849 | 165533 | 1082 | 165546 | 1030 | 157590 | 152 | 23256 | 263 | 40239 | 594 | 59400 |
| | 5 | 229 | 65832 | 630 | 132300 | 603 | 126630 | 96 | 20160 | 50 | 10500 | 387 | 38700 |
| | 8 | 266 | 57304 | 762 | 118872 | 744 | 116064 | 128 | 19968 | 70 | 10920 | 253 | 25300 |
| | 10 | 91 | 34565 | 286 | 78650 | 275 | 75625 | 63 | 17325 | 25 | 6875 | 266 | 26600 |
| | 15 | 250 | 46835 | 688 | 92880 | 698 | 94230 | 132 | 17820 | 85 | 11475 | 224 | 22400 |
| DTLZ7 | 3 | 600 | 127780 | 1134 | 173502 | 1124 | 171972 | 135 | 20655 | 242 | 37026 | 544 | 54400 |
| | 5 | 280 | 82057 | 618 | 129780 | 618 | 129780 | 92 | 19320 | 132 | 27720 | 398 | 39800 |
| | 8 | 327 | 71135 | 555 | 86580 | 636 | 99216 | 129 | 20124 | 89 | 13884 | 318 | 31800 |
| | 10 | 145 | 55524 | 273 | 75075 | 279 | 76725 | 65 | 17875 | 15 | 4125 | 297 | 29700 |
| | 15 | 301 | 56527 | 721 | 97335 | 667 | 90045 | 145 | 19575 | 67 | 9045 | 220 | 22000 |

As observed in Table 17, MOAVA based solution approach performed better in terms of IGD, GD, and HV. The performance measures have been drawn as histograms in the Fig 9 which shows the final solution set obtained for all algorithms with termination condition of 500 iteration. It is concluded that the proposed MOAVA provides very competitive results as compared to five well-known optimization algorithms in solving the investigated engineering real life applications.

## 6. Conclusion and future research directions

A novel many-objective African vulture optimization algorithm, named MaAVOA, is proposed in this paper. MaAVOA is an updated version of AVOA to handle the MaOPs. It

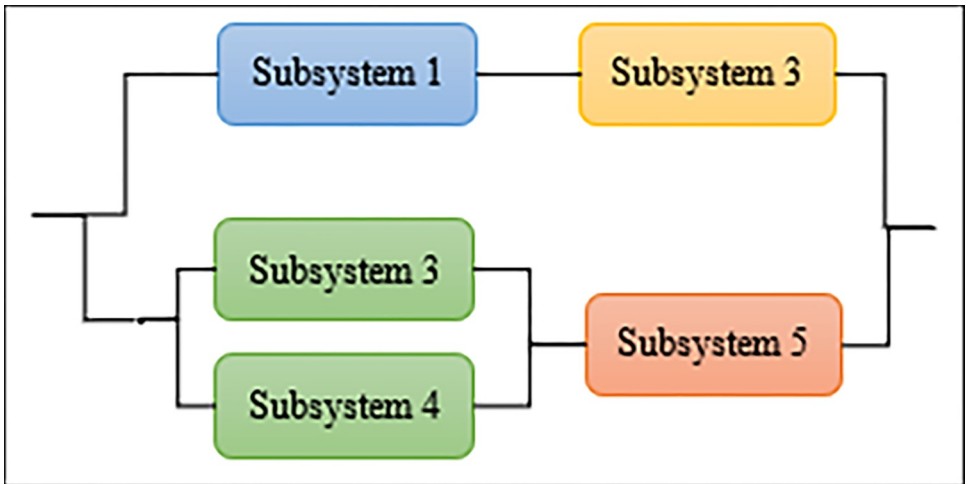

**Fig 6. Series-parallel system.**

**Table 14. Data used in series-parallel systems.**

| $i$ | $10^5 \alpha_i$ | $\beta_i$ | $w_i v_i^2$ | $w_i$ | $V$ | $C$ | $W$ |
|---|---|---|---|---|---|---|---|
| 1 | 2.5 | 1.5 | 2 | 3.5 | 180 | 175 | 100 |
| 2 | 1.45 | 1.5 | 4 | 4 | 180 | 175 | 100 |
| 3 | 0.541 | 1.5 | 5 | 4 | 180 | 175 | 100 |
| 4 | 0.541 | 1.5 | 8 | 3 | 180 | 175 | 100 |
| 5 | 2.1 | 1.5 | 4 | 4.5 | 180 | 175 | 100 |

**Table 15. The performance measures values of series-parallel system.**

| | MAAVOA | | | NSGAIII | | | U-NSGAIII | | | CTAEA | | | AGEMOEA | | |
|---|---|---|---|---|---|---|---|---|---|---|---|---|---|---|---|
| N_GEN | IGD | GD | HV | IGD | GD | HV | IGD | GD | HV | IGD | GD | HV | IGD | GD | HV |
| 250 | 8.90E-02 | 2.10E-01 | 9.01E-01 | 3.54E-02 | 6.94E-02 | 5.89E-01 | 3.71E-02 | 7.12E-02 | 5.90E-01 | 9.27E-02 | 1.27E-01 | 5.89E-01 | 1.04E-01 | 1.22E-01 | 5.82E-01 |
| 500 | 7.20E-02 | 2.05E-01 | 9.03E-01 | 4.35E-02 | 6.96E-02 | 5.90E-01 | 3.31E-02 | 6.69E-02 | 5.89E-01 | 9.78E-02 | 1.27E-01 | 5.89E-01 | 9.82E-02 | 1.19E-01 | 5.83E-01 |
| 1000 | 7.24E-02 | 2.08E-01 | 9.04E-01 | 3.49E-02 | 6.55E-02 | 5.90E-01 | 2.40E-02 | 6.05E-02 | 5.90E-01 | 1.09E-01 | 1.27E-01 | 5.89E-01 | 9.59E-02 | 1.12E-01 | 5.82E-01 |
| 2000 | 8.05E-02 | 2.13E-01 | 9.04E-01 | 3.46E-02 | 6.27E-02 | 5.90E-01 | 3.18E-02 | 6.44E-02 | 5.90E-01 | 1.01E-01 | 1.31E-01 | 5.89E-01 | 1.05E-01 | 1.14E-01 | 5.84E-01 |
| 4000 | 8.11E-02 | 2.12E-01 | 9.03E-01 | 2.84E-02 | 6.52E-02 | 5.90E-01 | 3.74E-02 | 7.00E-02 | 5.90E-01 | 9.99E-02 | 1.27E-01 | 5.89E-01 | 1.03E-01 | 1.20E-01 | 5.84E-01 |
| 5000 | 7.78E-02 | 2.14E-01 | 9.03E-01 | 3.05E-02 | 6.44E-02 | 5.90E-01 | 3.32E-02 | 6.80E-02 | 5.89E-01 | 1.06E-01 | 1.32E-01 | 5.89E-01 | 9.63E-02 | 1.14E-01 | 5.83E-01 |
| 10000 | 7.36E-02 | 2.18E-01 | 9.02E-01 | 2.84E-02 | 6.54E-02 | 5.90E-01 | 4.14E-02 | 7.20E-02 | 5.90E-01 | 9.25E-02 | 1.27E-01 | 5.89E-01 | 1.05E-01 | 1.19E-01 | 5.83E-01 |

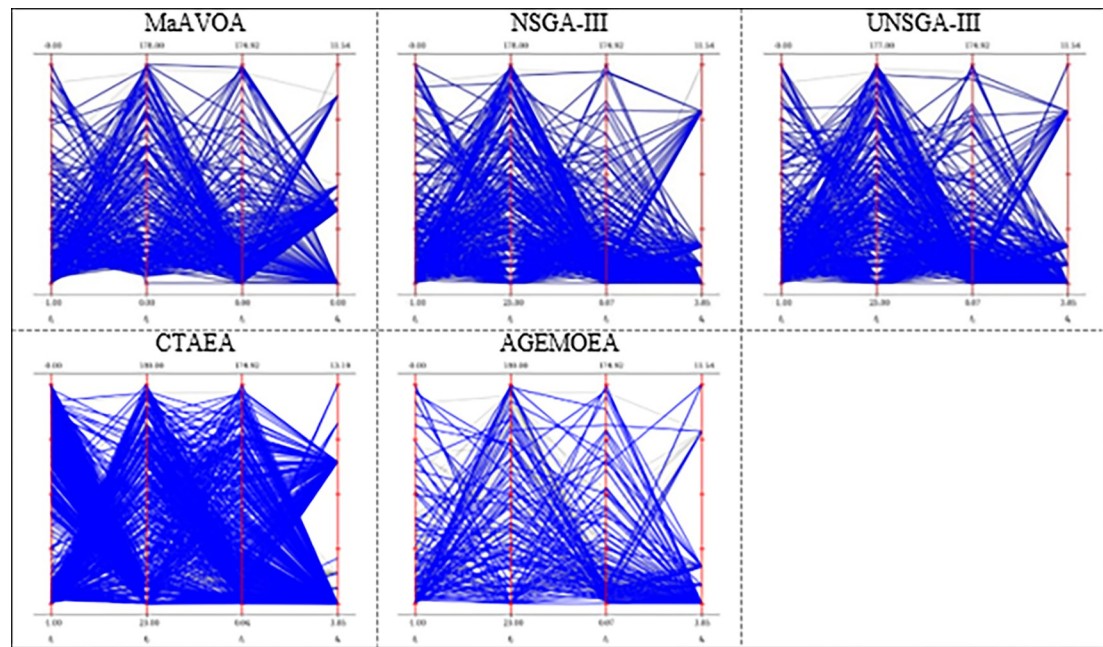

**Fig 7. Final solution set for a series-parallel system with five subsystems (termination condition is 500 iteration).**

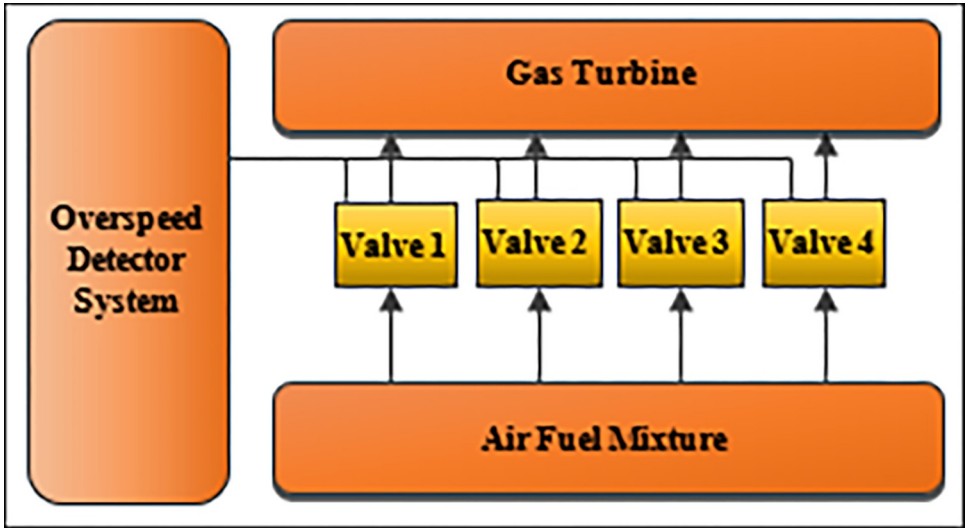

**Fig 8. Overspeed protection for gas turbine.**

**Table 16. Data used in Overspeed protection for gas turbine system.**

| $i$ | $10^5\alpha_i$ | $\beta_i$ | $v_i$ | $w_i$ | $V$ | $C$ | $W$ | $T$ |
|---|---|---|---|---|---|---|---|---|
| 1 | 1 | 1.5 | 1 | 6 | 250 | 400 | 500 | 1000h |
| 2 | 2.3 | 1.5 | 2 | 6 | 250 | 400 | 500 | 1000h |
| 3 | 0.3 | 1.5 | 3 | 8 | 250 | 400 | 500 | 1000h |
| 4 | 2.3 | 1.5 | 4 | 7 | 250 | 400 | 500 | 1000h |

**Table 17. The performance measures values of Overspeed protection for gas turbine.**

| N_GEN | MAAVOA | | | NSGAIII | | | U-NSGAIII | | | CTAEA | | | AGEMOEA | | |
|---|---|---|---|---|---|---|---|---|---|---|---|---|---|---|---|
| | IGD | GD | HV | IGD | GD | HV | IGD | GD | HV | IGD | GD | HV | IGD | GD | HV |
| 250 | 4.2E-02 | 6.1E-02 | 9.7E-02 | 5.2E-02 | 5.1E-02 | 1.5E-01 | 4.8E-02 | 6.0E-02 | 9.7E-02 | 1.5E-02 | 5.5E-02 | 1.4E-01 | 6.1E-02 | 6.9E-02 | 9.6E-02 |
| 500 | 4.3E-02 | 6.0E-02 | 9.7E-02 | 7.4E-03 | 3.1E-02 | 1.4E-01 | 4.5E-02 | 5.9E-02 | 9.7E-02 | 4.0E-02 | 5.4E-02 | 9.8E-02 | 6.5E-02 | 7.1E-02 | 9.6E-02 |
| 1000 | 4.3E-02 | 6.1E-02 | 9.7E-02 | 6.4E-03 | 3.0E-02 | 1.4E-01 | 5.5E-03 | 2.9E-02 | 1.4E-01 | 1.3E-02 | 5.3E-02 | 1.4E-01 | 6.5E-02 | 7.0E-02 | 9.6E-02 |
| 2000 | 4.5E-02 | 6.2E-02 | 9.7E-02 | 5.1E-02 | 5.1E-02 | 1.6E-01 | 1.4E-02 | 2.8E-02 | 1.4E-01 | 1.9E-02 | 7.0E-02 | 1.6E-01 | 6.2E-02 | 6.7E-02 | 9.5E-02 |
| 4000 | 4.6E-02 | 6.2E-02 | 9.7E-02 | 2.6E-03 | 2.7E-02 | 1.4E-01 | 6.1E-03 | 2.7E-02 | 1.4E-01 | 1.5E-02 | 5.8E-02 | 1.4E-01 | 6.7E-02 | 7.0E-02 | 9.6E-02 |
| 5000 | 4.5E-02 | 6.2E-02 | 9.7E-02 | 4.3E-02 | 4.8E-02 | 1.6E-01 | 1.4E-02 | 3.4E-02 | 1.4E-01 | 1.6E-02 | 6.9E-02 | 1.6E-01 | 6.2E-02 | 6.7E-02 | 9.6E-02 |
| 10000 | 5.0E-02 | 5.1E-02 | 1.6E-01 | 1.4E-02 | 3.2E-02 | 1.4E-01 | 4.4E-02 | 5.0E-02 | 1.6E-01 | 2.5E-02 | 7.5E-02 | 1.8E-01 | 5.7E-02 | 6.6E-02 | 9.6E-02 |

integrates a new social leader vultures selection process. In addition, an environmental selection mechanism based on the alternative pool was adapted to improve the selection pressure to maintain diversity for approximating different parts of the whole PF. An external Archive based on the FAM is established to save the best-nondominated solutions during the population evolution. Also, a RAS procedure is developed to improve the quality of archiving solutions and help reach out to the PF's missing areas that the vultures easily miss. The proposed MaAVOA was evaluated using well-known benchmark functions. Comparing the proposed MaAVOA results to five states of the art algorithms showed that MaAVOA outperformed the five algorithms in terms of IGD, GD, and HV in most of the benchmark test functions when all algorithms terminated according to several function evaluations or in case of terminating according to a maximum number of generations. To verify the performance of the proposed

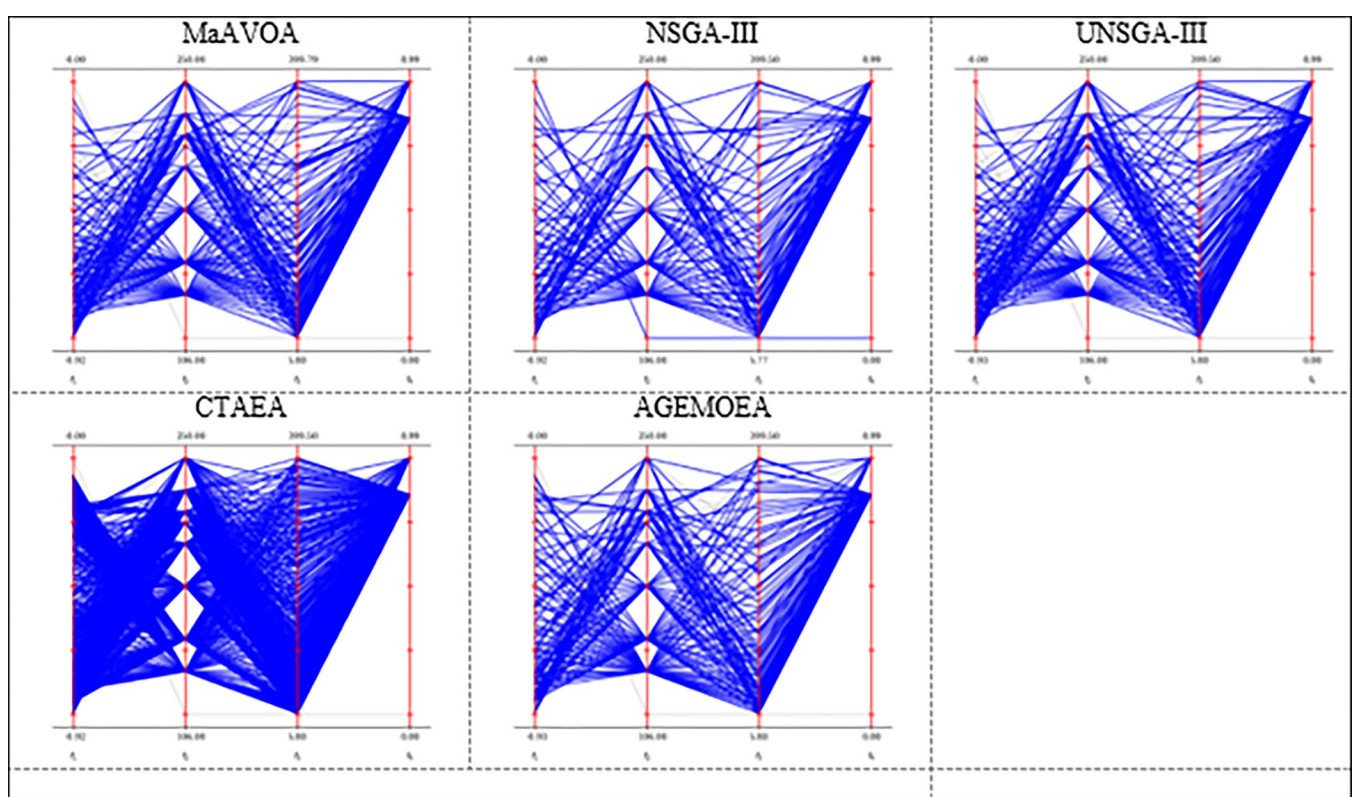

**Fig 9. Final solution set for overspeed protection for gas turbine problem (termination condition is 500 iteration).**

MaAVOA for real life many objectives' applications, it was applied and tested on two real-life engineering constrained problems. The findings show that among all the successful algorithms, MaAVOA has promising and competing performance.

There are many directions of research that can be recommended for future works to handle the limitations of the proposed work. The variation in operators of the proposed MaAVOA algorithm can motivate the future work to minimize the execution time of MaAVOA. Also, extending this algorithm to solve more constrained engineering many objective optimization problems can be seen as a future point for research. In addition, the computational time of the proposed algorithm is considered greater than both NSGAIII and UNSGAIII algorithms which can be considered as a future point for research. Furthermore, breaking out from the local optimum still difficult, so we suggest using a clustering strategy in the future to help.

## Author Contributions

**Conceptualization:** Aboul Ella Hassanien.

**Formal analysis:** Aboul Ella Hassanien.

**Methodology:** Heba Askr, M. A. Farag, Aboul Ella Hassanien.

**Project administration:** Václav Snášel.

**Software:** Heba Askr, M. A. Farag, Tamer Ahmed Farrag.

**Supervision:** Václav Snášel.

**Validation:** M. A. Farag, Tamer Ahmed Farrag.

**Visualization:** Tamer Ahmed Farrag.

**Writing – original draft:** Heba Askr, M. A. Farag, Tamer Ahmed Farrag.

**Writing – review & editing:** Aboul Ella Hassanien.

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
