## [Decision Letter · Decision Letter 0]

23 Jan 2023

PONE-D-22-28481Many-objective African vulture optimization algorithm for solving  the series-parallel system problem of real-life constrained engineeringPLOS ONE

Dear Dr. Hassanien,

Thank you for submitting your manuscript to PLOS ONE. After careful consideration, we feel that it has merit but does not fully meet PLOS ONE’s publication criteria as it currently stands. Therefore, we invite you to submit a revised version of the manuscript that addresses the points raised during the review process.

Please keep in mind that suggested citations is optional and any references amendments from the first version of the manuscript should be clearly and fully justified. 

We look forward to receiving your revised manuscript.

Kind regards,

Mohamed Kamel Riahi

Academic Editor

PLOS ONE

Journal Requirements:

"there is a statement of the funding on the paper"

4. We note that Figure (1) in your submission contain copyrighted images. All PLOS content is published under the Creative Commons Attribution License (CC BY 4.0), which means that the manuscript, images, and Supporting Information files will be freely available online, and any third party is permitted to access, download, copy, distribute, and use these materials in any way, even commercially, with proper attribution. For more information, see our copyright guidelines: http://journals.plos.org/plosone/s/licenses-and-copyright.

1. You may seek permission from the original copyright holder of Figure (1) to publish the content specifically under the CC BY 4.0 license. 

Additional Editor Comments (if provided):

Major revision is required.

Clarification of the point raised by reviewer is of utmost important, although, citing suggested articles has to be fully justified.

Reviewers' comments:

Reviewer's Responses to Questions

**Comments to the Author**

1. Is the manuscript technically sound, and do the data support the conclusions?

Reviewer #1: Yes

Reviewer #2: Yes

Reviewer #3: Yes

2. Has the statistical analysis been performed appropriately and rigorously? 

Reviewer #1: Yes

Reviewer #2: Yes

Reviewer #3: Yes

3. Have the authors made all data underlying the findings in their manuscript fully available?

Reviewer #1: Yes

Reviewer #2: Yes

Reviewer #3: Yes

4. Is the manuscript presented in an intelligible fashion and written in standard English?

Reviewer #1: Yes

Reviewer #2: Yes

Reviewer #3: Yes

5. Review Comments to the Author

Reviewer #1: This paper presents many-objective African vulture optimization algorithm for solving the series-parallel system problem of real-life constrained engineering. Before consider it for publication, the following comments should revised carefully:

1- The abstract should be improved including the novelties and software used in the presented work.

2- English typos and grammatical errors must be revised.

3- Some parameters are not defined.

4- Please double-check all the equations.

5- The parameters of used algorithms should be provided in table.

6- Different population and iteration sizes should be implemented to show the accuracy of presented technique.

7- The resolution of some Figs should be improved.

8- A recent optimization techniques should be implemented for comparison like YUKI algorithm, ….. .

9- The introduction should be improved including more references based on the most used optimization techniques in engineering applications such as damage identification, characterization, …. . Composite Structures 6 October 2022, 116272_Mechanics of Materials Volume 166, March 2022, 104200 _ Met. Mater. Int. 28, 370–384 (2022)_ Mechanics Volume 118, April 2022, 103213_Journal of Computational Science Volume 55, October 2021, 101451_Expert Systems with Applications Volume 186, 30 December 2021, 115669_Theoretical and Applied Fracture Mechanics Volume 107, June 2020, 102554_ Engineering Fracture Mechanics Volume 205, January 2019, Pages 285-300.

10- The conclusion should be improved and show the limitation of presented technique.

Reviewer #2: 1. The contribution is not stated clearly.

2. The choice of parameters used in the algorithm is not well justified.

3. A deep and detailed comparison with other methods is mandatory.

4. The authors claim that their method is faster and more efficient, but this is not rigorously demonstrated since it is applied just for a particular case.

5. What do you mean by experimental validation? Where the data exactly comes from, what is their reliability and accuracy for which model? Please address this important point seriously. Authors must cite the following papers;

Overall the quality of this paper is very good.

I recommend this paper. Authors must cite the following papers:

Singh, N., Hamid, Y., Juneja, S., Srivastava, G., Dhiman, G., Gadekallu, T. R., & Shah, M. A. (2023). Load balancing and service discovery using Docker Swarm for microservice based big data applications. Journal of Cloud Computing, 12(1), 1-9.

Nayak, J., Swapnarekha, H., Naik, B., Dhiman, G., & Vimal, S. (2022). 25 Years of Particle Swarm Optimization: Flourishing Voyage of Two Decades. Archives of Computational Methods in Engineering, 1-63.

Zhen, S., Surender, R., Dhiman, G., Rani, K. R., Ashifa, K. M., & Reegu, F. A. (2022). Intelligent-based ensemble deep learning model for security improvement in real-time wireless communication. Optik, 271, 170123.

Singh, Shailendra Pratap, Wattana Viriyasitavat, Sapna Juneja, Hani Alshahrani, Asadullah Shaikh, Gaurav Dhiman, Aman Singh, and Amandeep Kaur. "Dual adaption based evolutionary algorithm for optimized the smart healthcare communication service of the Internet of Things in smart city." Physical Communication 55 (2022): 101893.

Rani, S., Babbar, H., Srivastava, G., Gadekallu, T. R., & Dhiman, G. (2022). Security Framework for Internet of Things based Software Defined Networks using Blockchain. IEEE Internet of Things Journal.

Singamaneni, K. K., Nauman, A., Juneja, S., Dhiman, G., Viriyasitavat, W., Hamid, Y., & Anajemba, J. H. (2022). An Efficient Hybrid QHCP-ABE Model to Improve Cloud Data Integrity and Confidentiality. Electronics, 11(21), 3510.

Shukla, Surendra Kumar, Bhaskar Pant, Wattana Viriyasitavat, Devvret Verma, Sandeep Kautish, Gaurav Dhiman, Amandeep Kaur, Kannan Srihari, and Sachi Nandan Mohanty. "An integration of autonomic computing with multicore systems for performance optimization in Industrial Internet of Things." IET Communications (2022).

Singh, S. P., Dhiman, G., Viriyasitavat, W., & Kautish, S. (2022). A Novel Multi-Objective Optimization Based Evolutionary Algorithm for Optimize the Services of Internet of Everything. IEEE Access, 10, 106798-106811.

Alrashed, Fahad Abdulaziz, Abdulrahman M. Alsubiheen, Hessah Alshammari, Sarah Ismail Mazi, Sara Abou Al-Saud, Samha Alayoubi, Shaji John Kachanathu et al. "Stress, Anxiety, and Depression in Pre-Clinical Medical Students: Prevalence and Association with Sleep Disorders." Sustainability 14, no. 18 (2022): 11320.

Singamaneni, K. K., Dhiman, G., Juneja, S., Muhammad, G., AlQahtani, S. A., & Zaki, J. (2022). A novel QKD approach to enhance IIOT privacy and computational knacks. Sensors, 22(18), 6741.

Ahmad, F., Shahid, M., Alam, M., Ashraf, Z., Sajid, M., Kotecha, K., & Dhiman, G. (2022). Levelized Multiple Workflow Allocation Strategy under Precedence Constraints with Task Merging in IaaS Cloud Environment. IEEE Access, 10, 92809-92827.

Kumar, R., & Dhiman, G. (2021). A Comparative Study of Fuzzy Optimization through Fuzzy Number. International Journal of Modern Research, 1, 1-14.

Chatterjee, I. (2021). Artificial Intelligence and Patentability: Review and Discussions. International Journal of Modern Research, 1, 15-21.

Vaishnav, P.K., Sharma, S., & Sharma, P. (2021). Analytical Review Analysis for Screening COVID-19. International Journal of Modern Research, 1, 22-29.

Gupta, V. K., Shukla, S. K., & Rawat, R. S. (2022). Crime tracking system and people’s safety in India using machine learning approaches. International Journal of Modern Research, 2(1), 1-7.

Sharma, T., Nair, R., & Gomathi, S. (2022). Breast Cancer Image Classification using Transfer Learning and Convolutional Neural Network. International Journal of Modern Research, 2(1), 8-16.

Shukla, S. K., Gupta, V. K., Joshi, K., Gupta, A., & Singh, M. K. (2022). Self-aware Execution Environment Model (SAE2) for the Performance Improvement of Multicore Systems. International Journal of Modern Research, 2(1), 17-27

6. The authors should provide other applications of the proposed algorithm.

7. To demonstrate the effectiveness of the proposed algorithm a real experimental validation is mandatory for a rigorous and accurate comparison and validation.

8. Please discuss the performance of the technique for real-time applications?

9. Specify the limitations and drawbacks of the proposed method.

10. The conclusion must be rewritten.

Reviewer #3: Suggest the authors highlight the main contributions of their work.

A clear statement about contributions should be given in the Introduction section.

Suggest the authors have a solid statement about the proposed method by providing a motivational example.

Suggest the authors add more experimental results.

The abstract needs to update.

The experimental results in this paper are also somewhat weak so need to verify your proposed method in the present form.

The contribution of the paper can be neatly elaborated by including the novelty of the paper and the major difference from available models.

The author's research is very meaningful, and I hope you will explain the experiment further.

The technical details make it much more intelligible, so please provide some strong technical details in the main methodology.

The researchers are recommended to revise the result section because the comparative results with the existing approaches are not explained briefly.

The conclusion should indicate the experimental evaluation's implications and include some obtained values to point out the superiority.

6. PLOS authors have the option to publish the peer review history of their article (what does this mean?). If published, this will include your full peer review and any attached files.

Reviewer #1: No

Reviewer #2: No

Reviewer #3: **Yes: **Dr B Santhosh Kumar

---

## [Decision Letter · Decision Letter 1]

23 Mar 2023

Many-Objective African Vulture Optimization Algorithm: A Novel Approach for Many-Objective Problems

PONE-D-22-28481R1

Dear Dr. Hassanien,

We’re pleased to inform you that your manuscript has been judged scientifically suitable for publication and will be formally accepted for publication once it meets all outstanding technical requirements.

Kind regards,

Mohamed Kamel Riahi

Academic Editor

PLOS ONE

Additional Editor Comments (optional):

Reviewers' comments:

Reviewer's Responses to Questions

**Comments to the Author**

1. If the authors have adequately addressed your comments raised in a previous round of review and you feel that this manuscript is now acceptable for publication, you may indicate that here to bypass the “Comments to the Author” section, enter your conflict of interest statement in the “Confidential to Editor” section, and submit your "Accept" recommendation.

Reviewer #1: (No Response)

Reviewer #2: (No Response)

Reviewer #3: All comments have been addressed

2. Is the manuscript technically sound, and do the data support the conclusions?

Reviewer #1: No

Reviewer #2: (No Response)

Reviewer #3: Yes

3. Has the statistical analysis been performed appropriately and rigorously? 

Reviewer #1: No

Reviewer #2: (No Response)

Reviewer #3: Yes

4. Have the authors made all data underlying the findings in their manuscript fully available?

Reviewer #1: (No Response)

Reviewer #2: (No Response)

Reviewer #3: Yes

5. Is the manuscript presented in an intelligible fashion and written in standard English?

Reviewer #1: Yes

Reviewer #2: (No Response)

Reviewer #3: Yes

6. Review Comments to the Author

Reviewer #1: (No Response)

Reviewer #2: I recommend this work for future publication in this work. Hence i recommend to accept this work for this journal.

Reviewer #3: The whole article is properly written understandably. Moreover, this article sounds well with various aspects in this research area and the involvement of this work is appreciable.

7. PLOS authors have the option to publish the peer review history of their article (what does this mean?). If published, this will include your full peer review and any attached files.

Reviewer #1: **Yes: **Samir Khatir

Reviewer #2: **Yes: **Gaurav Dhiman

Reviewer #3: **Yes: **Dr B Santhosh Kumar

---

## [Editor Report · Acceptance letter]

29 Mar 2023

PONE-D-22-28481R1 

Many-Objective African Vulture Optimization Algorithm: A Novel Approach for Many-Objective Problems 

Dear Dr. Hassanien:

I'm pleased to inform you that your manuscript has been deemed suitable for publication in PLOS ONE. Congratulations! Your manuscript is now with our production department. 

Kind regards, 

on behalf of

Dr. Mohamed Kamel Riahi 

Academic Editor

PLOS ONE